# HybridLinker: Topology-Guided Posterior Sampling for Enhanced Diversity and Validity in 3D Molecular Linker Generation

## Abstract

Linker generation is critical in drug discovery applications such as lead optimization and PROTAC design, where molecular fragments are assembled into diverse drug candidates via molecular linker. Existing methods fall into point cloud-free and point cloud-aware categories based on their use of fragments' 3D poses alongside their topologies in sampling the linker's topology. Point cloud-free models prioritize sample diversity but suffer from lower validity due to overlooking fragments' spatial constraints, while point cloud-aware models ensure higher validity but restrict diversity by enforcing strict spatial constraints. To overcome these trade-offs without additional training, we propose HybridLinker, a framework that enhances point cloud-aware inference by providing diverse bonding topologies from a pretrained point cloud-free model as guidance. At its core, we propose LinkerDPS, the first diffusion posterior sampling (DPS) method operating across point cloud-free and point cloud-aware spaces, bridging molecular topology with 3D point clouds via an energy-inspired function. By transferring the diverse sampling distribution of point cloud-free models into the point cloud-aware distribution, HybridLinker significantly surpasses baselines, improving both validity and diversity in foundational molecular design and applied drug optimization tasks, establishing a new DPS framework in the molecular domains beyond imaging.

## 1 Introduction

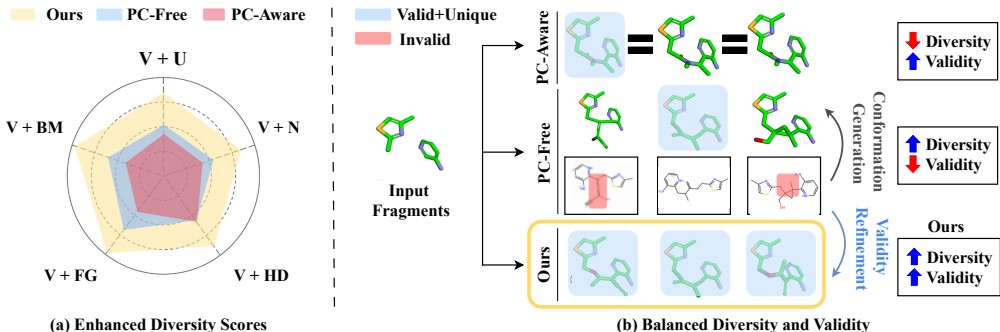

(a) Enhanced Diversity Scores      (b) Balanced Diversity and Validity

Figure 1: (a) Qualitative comparison of our method and existing pipelines on diversity of valid molecule samples measured on five metrics. (b) Trade-off between diversity and validity in baseline models and our hybrid approach overcoming these limitations.

Fragment-based drug discovery strategically assembles molecular fragments to create stable and effective drug candidates. A key challenge is linker generation, which involves designing molecular linkers that connect fragments of fixed 3D poses while ensuring their validity. In this work, we categorize existing linker generation models into Point Cloud-Free (PC-Free) and Point Cloud-Aware (PC-Aware) models, distinguished by whether they account for the 3D conformation (3D poses) of fragments when determining the bonding topology of the complete molecule. Furthermore, through a fair comparison, we highlight that achieving a balance between diversity (exploring novel topological spaces) and validity (ensuring spatial consistency between fragments and the linker) remains a significant challenge in current linker design methodologies.

Figure 2: Comparison of generation pipelines for PC-Free and PC-Aware models. HybridLinker is designed to leverage the strengths of both approaches, inheriting high diversity from the PC-Free model and high validity from the PC-Aware model.

**Point Cloud-Free Models (↑ Diversity ↓ Validity).** These models, such as FFLOM(Jin et al., 2023) and DeLinker(Imrie et al., 2020), generate diverse bonding topologies based on fragment connectivity while ignoring the 3D conformations of the given fragments. Subsequently, they predict 3D conformations that reflect the predefined fragment geometries and the sampled topology using conformation generators(Riniker & Landrum, 2015; Xu et al., 2022; 2021). By excluding 3D fragment conformations during topology generation, these methods increase entropy in the sampling distribution, thereby enhancing diversity. However, the sampled topology often fails to produce a ligand conformation that aligns well with the predefined fragment geometry, resulting in invalid, high-energy molecules

**Point Cloud-Aware Models (↓ Diversity ↑ Validity).** DiffLinker(Igashov et al., 2024) and 3DLinker(Huang et al., 2022) achieve high validity by incorporating fragments' 3D poses when sampling the linker's bonding topology, ensuring spatial alignment between fragments and generated linkers. They achieve this by performing generation in point cloud space, where the 3D poses of fragments are explicitly encoded as atomic coordinates, and the linker's topology is derived by from the linker's point cloud generated to align with the fragments' atomic coordinates. However, their strict spatial constraints lower sampling entropy, limiting the exploration of diverse topologies. This constrained search space increases the risk of overfitting, making it challenging to generate topologically diverse drug candidates.

To overcome the diversity-validity trade-off, we propose **HybridLinker**, a framework that integrates the strengths of PC-Free and PC-Aware models. By guiding a PC-Aware model with bonding topologies sampled from a PC-Free model, HybridLinker enhances diversity while preserving validity. At its core, we introduce LinkerDPS, the first diffusion posterior sampling (DPS) method that bridges molecular topology and point cloud space via an energy-inspired function. This approach transfers the highly diverse samples of PC-Free models into the validity-focused distribution of PC-Aware models, achieving a balanced trade-off between diversity and validity. Figure 1 illustrates how HybridLinker surpasses existing methods.

We evaluate HybridLinker on the ZINC(Irwin et al., 2020) test dataset, a standard benchmark in drug discovery, demonstrating its ability to generate diverse and valid molecules from fragment inputs. By leveraging training-free cooperation between PC-Free and PC-Aware models, HybridLinker surpasses existing methods in diversity of valid molecules and its enhanced diversity also drives superior performance in drug-likeness optimization task, highlighting its potential as a foundational model. Moreover, HybridLinker's success validates LinkerDPS, showcasing its versatility across various domains.

Our contributions can be summarized as follows:

- We are the first to highlight the trade-off between diversity and validity in a fair comparison of Point Cloud-Free and Point Cloud-Aware linker generation models.

- We present HybridLinker, a simple yet effective framework that integrates pretrained Point Cloud-Free and Point Cloud-Aware linker generation models within a two-step generation pipeline, enabling training-free inference inheriting their strengths.

Table 1: Trade-off between diversity and validity in PC-Free and PC-Aware models. The **F/A** column indicates whether the model belongs to PC-Free (F) or PC-Aware (A). Uniqueness is a metric that measures the diversity of samples, while V + U represents uniqueness counting only valid molecules.

| Method | F / A | Uniquess (↑) | Validity (↑) | V + U (↑) |
|--------|-------|--------------|--------------|-----------|
| FFLOM | F | **0.840** (High) | 0.370 (Low) | 0.313 |
| DeLinker | F | 0.638 (High) | 0.575 (Low) | **0.386** |
| DiffLinker | A | 0.349 (Low) | **0.711** (High) | 0.269 |
| 3DLinker | A | 0.443 (Low) | 0.654 (High) | 0.326 |

- We introduce LinkerDPS, the first DPS method beyond the image domain, operating across molecular topology and point cloud spaces. It bridges these domains through an energy-inspired cross-domain function, enabling effective topology-guided molecular point cloud generation. Its cross-domain guidance overcomes challenges in point cloud space by leveraging topology space as an intermediary, offering a wide range of potential applications.

- We evaluate HybridLinker across both the fundamental task of diverse drug-candidates sampling and the application-driven task of drug-likeness optimization, demonstrating its potential as a foundation model. Furthermore, validated perfomance of LinkerDPS in linker generation show its potential for broader applications, particularly in challenging point cloud tasks benefiting from topological guidance.

## 2 BACKGROUND

In the appendix, we provide a table of notations (Appendix A), a summary of relevant tasks in molecule generation (Appendix P), and an extended definition of molecular validity that incorporates 3D conformation alongside topological rules (Appendix C).

### 2.1 MOLECULAR REPRESENTATIONS

We provide a detailed description of molecular representations in Appendix B. Briefly, we represent a molecule as a 3D graph $G = (V, E, R)$, where $V \in \{0, 1\}^{N \times A}$ denotes the one-hot encoded atom types for $N$ atoms and $A$ atom types, $E \in \{0, 1\}^{N \times N \times B}$ represents the presence of $B$ bond types between atom pairs, and $R \in \mathbb{R}^{N \times 3}$ specifies the 3D coordinates (i.e., 3D pose) of the atoms. We further define two key sub-representations: the **bonding topology** $\mathcal{T} = (V, E)$, which captures atomic connectivity, and the **point cloud representation** $(V, R)$, which captures the spatial configuration of atoms. A 3D molecular graph can thus be equivalently expressed as $G = (\mathcal{T}, R)$, combining both topological and geometric information. We denote by $\mathbb{P}_G = \mathbb{P}_{\mathcal{T}, R}$ the distribution (i.e., dataset) of valid 3D molecular graphs, viewed as the joint distribution over bonding topologies and 3D conformations. In prior works on molecular modeling (Hoogeboom et al., 2022; Igashov et al., 2024; Huang et al., 2022; Jin et al., 2023), the choice of molecular representation varies depending on task objectives and modeling efficiency.

**Point Cloud Representation for 3D Diffusion Models.** In recent 3D molecule diffusion models (Hoogeboom et al., 2022; Igashov et al., 2024; Corso et al., 2023b; Schneuing et al., 2024; Corso et al., 2023a), the one-hot encoded atom types $V \in \{0, 1\}^{N \times A}$ and atomic coordinates $R \in \mathbb{R}^{N \times 3}$ are flattened and embedded into a continuous space $\mathbb{R}^d$, where $d = N \times (A + 3)$, combining both atom identity and spatial information. The diffusion process operates in this continuous space. After sampling, discrete atom types are recovered via $\mathrm{argmax}$ over $V$, and chemical bonds $E$ are predicted post hoc using a bond predictor $\mathcal{E}$, resulting in the final molecular graph $G = (V, E, R)$.

### 2.2 PROBLEM SETUP:LINKER GENERATION

Linker generation (Shuker et al., 1996) is a fundamental approach for high-affinity drug design, starting from fragments of a reference drug that interact with the target protein. The task is defined as generating submolecules, referred to as linkers, which connect the two fragments into a complete molecule. The 3D poses of the fragments are preserved as in the reference drug to maintain their spatial compatibility with the binding pocket, thereby contributing to the high affinity of the generated molecules.

Figure 3: Problem setup for linker generation with its context in drug discovery, illustrating the fragments $G_1$ and $G_2$ embedded in a candidate molecule $G'$. $N_{\text{ref}}$ is the size of reference molecule.

**Input.** Suppose we are given two molecule fragments $G_1 = (\mathcal{T}_1, R_1)$ and $G_2 = (\mathcal{T}_2, R_2)$, where $G_1$ and $G_2$ are subgraphs of a reference molecule $G_{\text{ref}} = (\mathcal{T}_{\text{ref}}, R_{\text{ref}})$. The reference molecule $G_{\text{ref}}$, which contains $N_{\text{ref}}$ atoms, is chemically valid, and belongs to a molecular graph dataset. Further, we set $\mathcal{T}_{\text{cond}} = \mathcal{T}_1 \cup \mathcal{T}_2$, and $R_{\text{cond}} = R_1 \cup R_2$.

**Output.** Our task is to generate new complete 3D molecular graphs $G' = (\mathcal{T}', R')$, by sampling from the following conditional distribution[1] (Figure 3):

$$G' \sim \mathbb{P}_G(G \mid \underbrace{G_1, G_2 \subset G, N_{\text{ref}} \text{ atoms}}_{\mathcal{F} = \mathcal{T}_{\text{cond}} \cap R_{\text{cond}}: \text{ fragment conditions}}) \tag{1}$$

For simplicity, we use the shorthand $\mathbb{P}_G(G \mid \mathcal{F})$ to denote this joint distribution which captures the topological ($\mathcal{T}_{\text{cond}}$) and geometric ($R_{\text{cond}}$) constraints imposed by a fragment condition $\mathcal{F}$. We analogously write the marginal distributions of $\mathcal{T}, R$ as $\mathbb{P}_{\mathcal{T}}$ and $\mathbb{P}_R$, respectively.

## 2.3 PC-Free and PC-Aware Approaches

Existing works in linker generation divide into two approaches: Point Cloud Free (PC-Free) and Point Cloud Aware (PC-Aware), based on the utilization of fragments' 3D poses $R_{\text{cond}}$ when sampling the topology of the output molecule $\mathcal{T}$. To clarify, PC-Free models use the following approximation:

$$\mathbb{P}_{\mathcal{T}}(\mathcal{T} \mid \underbrace{\mathcal{T}_{\text{cond}}, R_{\text{cond}}}_{\mathcal{F}}) \approx \mathbb{P}_{\mathcal{T}}(\mathcal{T} \mid \mathcal{T}_{\text{cond}}), \tag{2}$$

and the probability distribution of $\mathcal{T}$ in two approaches' samples are distinguished as following:

$$\begin{aligned} \text{PC-Free} &: \mathbb{P}_{\mathcal{T} \mid \mathcal{T}_{\text{cond}}}(\mathcal{T} \mid \mathcal{T}_{\text{cond}}) \\ \text{PC-Aware} &: \mathbb{P}_{\mathcal{T} \mid \mathcal{T}_{\text{cond}}, R_{\text{cond}}}(\mathcal{T} \mid \mathcal{T}_{\text{cond}}, \boxed{R_{\text{cond}}}) \end{aligned} \tag{3}$$

In the following, we explain each approach in detail with related models.

**Point Cloud Free (PC-Free) Approach.** PC-Free approaches (Jin et al., 2023; Imrie et al., 2020) model the conditional distribution as in equation 4, sequentially sampling the bonding topology $\mathcal{T}$ conditioned on **only the fragments' bonding topology** $\mathcal{T}_{\text{cond}}$ and then the 3D conformation $R$.

$$\begin{aligned} \mathbb{P}_G(G \mid \mathcal{F}) &\overset{\text{Bayes}}{=} \mathbb{P}_{\mathcal{T}}(\mathcal{T} \mid \boxed{\mathcal{F}}) \cdot \mathbb{P}_{R \mid \mathcal{T}}(R \mid \mathcal{T}, \mathcal{F}) \\ &\overset{equation\ 2}{\approx} \mathbb{P}_{\mathcal{T}}(\mathcal{T} \mid \boxed{\mathcal{T}_{\text{cond}}}) \cdot \mathbb{P}_{R \mid \mathcal{T}}(R \mid \mathcal{T}, \mathcal{F}) \\ &=: \mathbb{P}_{\text{2D}}(G \mid \mathcal{F}) \end{aligned} \tag{4}$$

However, the approximation in equation 2, which assumes that the linker's topology is independent of $R_{\text{cond}}$, is often inaccurate. In reality, the topology of the linker generally depends on $R_{\text{cond}}$, as its topology-driven conformation must align with $R_{\text{cond}}$ to ensure energetic stability. To emphasize the distinction between the true distribution $\mathbb{P}_G$ and its approximation, we denote the latter as $\mathbb{P}_{\text{2D}}$, referring to it as the **surrogate distribution**.

---

[1]We write $G_1 \subset G'$ to denote that $G'$ contains $G_1$ as its subgraph, meaning that $V_1 \subset V', E_1 \subset E', R_1 \subset R'$. We analogously write $\mathcal{T}_1 \subset \mathcal{T}$ to mean $V_1 \subset V', E_1 \subset E'$.

Figure 4: Conceptual comparison of sampling distributions in Diffusion-based PC-Aware, PC-Free, and HybridLinker models. (a) Diffusion-based PC-Aware models focus on validity but suffer from low diversity due to spatial constraints. (b) PC-Free models explore a broad molecular space but often generate invalid molecules. (c) HybridLinker leverages LinkerDPS to balance diversity and validity, enhancing exploration while maintaining correctness.

PC-Free models sample from equation 4 by using a neural network to approximate $\mathbb{P}_{\mathcal{T}}(\mathcal{T} \mid \mathcal{T}_{\text{cond}})$ and an off-the-shelf conformation generator $\mathcal{R}$, i.e., RDkit (Landrum, 2022), which produces the 3D conformation of the molecule including condition of $R_1$ and $R_2$ as $R' = \mathcal{R}(\mathcal{T}', R_1, R_2) \in \mathbb{R}^{N \times 3}$.

**Point Cloud Aware (PC-Aware) Approach.**   In contrast, PC-Aware models like DiffLinker (Igashov et al., 2024) and 3DLinker (Huang et al., 2022) learn the distribution of bonding topology conditioned on both 3D poses and bonding topology of fragments, as in equation 3. In particular, DiffLinker and 3DLinker incorporate atom-wise distances between linker atoms and fragments to account for $R_{\text{cond}}$ in determining both topology and conformation of linker atoms, thereby resulting in a non-approximated sampling framework equation 5.

$$\mathbb{P}_G(G \mid \mathcal{F}) = \underbrace{\mathbb{P}_{V,R}(V, R \mid \mathcal{F})}_{\text{Generative Model}} \cdot \underbrace{\mathbb{P}_{E|V,R}(E \mid \mathcal{F}, V, R)}_{E = \mathcal{E}(\mathcal{F}, V, R)} \tag{5}$$

This formulation allows generative models to focus on sampling atomic positions while leveraging external bond inference model $\mathcal{E}$, i.e., Open Babel (O'Boyle et al., 2011).

## 2.4   Trade-Off in Diversity and Validity

In generating drug candidates, attaining both high validity and diveristy is crucial. However, due to different sampling strategy, PC-Free and PC-Aware models exhibit an inverse relationship between diversity and validity. Specifically, PC-Free models demonstrate **high diversity but low validity**, whereas PC-Aware models achieve **high validity but low diversity**.

**Diversity**   PC-Free models generate $\mathcal{T}$ without conditioning on $R_{\text{cond}}$, leading to greater diversity compared to PC-Aware models, which incorporate $R_{\text{cond}}$ in sampling $\mathcal{T}$. This follows from the general principle that conditioning reduces entropy,

$$H(\mathcal{T} \mid \mathcal{T}_{\text{cond}}, R_{\text{cond}}) \leq H(\mathcal{T} \mid \mathcal{T}_{\text{cond}}). \tag{6}$$

**Validity**   PC-Free models face validity issues due to the inaccurate assumptions equation 2 in equation 4. As described in equation 3, since the 3D conformations of fragments, $R_1$ and $R_2$, are ignored when determining the bonding topology $\mathcal{T}'$, even a model that perfectly learns the target distribution $\mathbb{P}_{\text{2D}}$ will still deviate from the true molecular distribution $\mathbb{P}_G$. In contrast, PC-Aware models inherently overcome this limitation by explicitly training on the linker generation task in equation 5.

## 2.5   Diffusion-based PC-Aware Model

Following their success in a wide range of molecular generation tasks (Hoogeboom et al., 2022; Igashov et al., 2024; Corso et al., 2023b), diffusion-based models have recently become the baseline for PC-Aware linker generation (Igashov et al., 2024). It leverages the reverse SDE to iteratively denoise a randomly sampled $(v_T, r_T) \sim \mathcal{N}(0, I)$ into a valid point cloud $(v_0, r_0) = (V, R)$, conditioned on fragment input $\mathcal{F}$, over $T$ timesteps. Specifically, the process is formulated as

$$\begin{cases} (v_t, r_t) \sim \mathcal{N}(0, I) \\ d(v_t, r_t) = \left[ f(v_t, r_t, t) - g^2(t) \nabla_{v_t, r_t} \log p_t(v_t, r_t \mid \mathcal{F}) \right] dt + g(t) d\bar{W}_t \end{cases} \tag{7}$$

where $\bar{W}_t$ denotes the reverse Wiener process. The drift coefficient is defined as $f(v_t, r_t, t) = -\frac{1}{2}\beta_t \cdot (v_t, r_t)$, and the diffusion coefficient as $g(t) = \sqrt{\beta_t}$, with $\beta_t$ being a predefined noise schedule. The term $\nabla_{v_t, r_t} \log p_t(v_t, r_t)$, referred to as the score, is estimated using a trained network $s_{\theta^*}$. The final denoised graph $x_0$ is converted into a one-hot encoded molecular point cloud using `argmax`, followed by a post-hoc bond predictor, as described in Section 2.1.

# 3 METHOD

Rather than directly sampling $G' \sim \mathbb{P}_G$ in a single step, we propose a hybrid two-step approach that leverages both PC-Free and PC-Aware linker generation models.

## 3.1 HYBRID APPROACH

We develop a straightforward and intuitive pipeline that integrates two pretrained models, leveraging a novel theoretical methodology to execute the process. To adopt two types of models, we rewrite our sampling as two-step pipeline given a fragment condition $\mathcal{F} = \mathcal{T}_{\text{cond}} \cap R_{\text{cond}}$ as $\mathbb{P}_{G,\tilde{G}}(G, \tilde{G} \mid \mathcal{F}) = \mathbb{P}_{\tilde{G}}(\tilde{G} \mid \mathcal{F}) \cdot \mathbb{P}_{G|\tilde{G}}(G \mid \tilde{G}, \mathcal{F})$, where we first sample surrogate molecule $\tilde{G}$ to support sampling of $G$.

Now, we plug-in the PC-Free and PC-Aware models to implement the two steps pipeline. In the first stage, we set $\mathbb{P}_{\tilde{G}}$ as the PC-Free distribution $\mathbb{P}_{\text{2D}}$, which captures the topological diversity of $\mathbb{P}_G$ but also includes invalid samples. Correspondingly, $\mathbb{P}_{G|\tilde{G}}$ in the second stage becomes sampling of valid molecule equal or similar to $\tilde{G}$. To clarify, if $\tilde{G}$ is invalid, we run a molecular refinement process to obtain a similar but valid molecule; otherwise, we set $G$ to be $\tilde{G}$. To implement the molecular refinement process, we perform posterior sampling using the validity-focused prior distribution of $G$, learned by the PC-Aware model, and our likelihood of $\tilde{G}$, which favors similarity with $G$. To summarize, we implement the pipeline as equation 8.

$$\mathbb{P}_{G,\text{2D}}(G, \tilde{G} \mid \mathcal{F}) = \underbrace{\mathbb{P}_{\text{2D}}(\tilde{G} \mid \mathcal{F})}_{\text{Step 1: PC-Free}} \cdot \underbrace{\mathbb{P}_{G|\text{2D}}(G \mid \tilde{G}, \mathcal{F})}_{\text{Step 2: PC-Aware}} \tag{8}$$

At a high level, our implementation can be viewed as transferring the high-entropy of the PC-Free distribution into the validity-focused PC-Aware distribution, leading to a coordinated distribution that ensures both high diversity and validity of $G$. Specifically, our process produces a tuple comprising a valid 3D molecule sample $G \sim \mathbb{P}_G$ and a surrogate $\tilde{G} \sim \mathbb{P}_{\text{2D}}$. We describe the algorithmic difference of HybridLinker to PC-Aware model and PC-Free model in Appendix E. Similary, Figure 4 visualizes and compares the sampling distribution of PC-Aware, PC-Free, and HybridLinker.

## 3.2 MOTIVATION FOR A HYBRID APPROACH

**Experimental Comparison** To validate this trade-off, we evaluate PC-Free and PC-Aware models on the ZINC dataset using standard metrics: **Validity** for validity, **Uniqueness** for diversity and **V + U** for diversity of valid molecules. The experimental setup follows Section 4.1. As shown in Table 1, PC-Free models excel in diversity, while PC-Aware models perform best in validity, confirming our hypothesis.

As observed in V + U score, the trade-off between diversity and validity significantly limits the practical utility of existing linker generation models. To overcome this limitation, we propose a **hybrid approach** that integrates the strengths of both PC-Free and PC-Aware models without requiring any training. Our hybrid pipeline leverages pretrained PC-Free and PC-Aware models cooperatively to achieve both high diversity and validity. Figure 2 illustrates the generation processes of PC-Free and PC-Aware models, along with goal of our proposed hybrid strategy of inheriting strength of two models.

## 3.3 POSTERIOR SAMPLING FROM $\mathbb{P}_{G|\text{2D}}$ VIA LINKERDPS

The challenge in sampling $\mathbb{P}_{G|\text{2D}}$ lies in refining an invalid surrogate molecule into a valid one, which corresponds to the inverse problem(Gutha et al., 2024; Yang et al., 2024; Chung et al., 2024a) in the molecular domain. Since this area remains unexplored, we introduce LinkerDPS, the first DPS(Chung

Table 2: Comparison of linker generation models across diversity and validity metrics. The F/A column indicates whether the model belongs to a PC-Free (F), PC-Aware (A), or hybrid (F + A) category. The "Diversity w/ Validity" columns quantify the diversity of molecules that meet validity criteria and underscore HybridLinker's ability to sample a broad range of valid drug candidates. We also report the supporting metrics that independently measure validity and diversity without considering validity (i.e., only satisfying topological valence rules) in the columns named "Validity" and "Diversity w/o Validity," respectively.

| | F/A | • Diversity w/ Validity | | | | | Validity (%) | Diversity w/o Validity | |
| | | V+U (%) | V+N (%) | V+HD | V+FG | V+BM | | Uniqueness (%) | Novelty (%) |
|---|---|---|---|---|---|---|---|---|---|
| FFLOM | F | 31.25 | 31.32 | 6.38 | 16.37 | 14.48 | 37.07 | **84.04** | **58.29** |
| DeLinker | F | 38.57 | 38.53 | 7.32 | 16.20 | 17.33 | 57.44 | 63.78 | 31.63 |
| DiffLinker | A | 26.90 | 26.90 | 5.08 | 11.37 | 10.91 | 71.08 | 34.94 | 16.32 |
| 3DLinker | A | 32.61 | 32.64 | 6.14 | 15.40 | 14.23 | 65.31 | 44.31 | 21.83 |
| HybridLinker (FFLOM) | F+A | **55.10** | 55.09 | **10.81** | **23.92** | **24.59** | 69.03 | 68.39 | 44.67 |
| HybridLinker (DeLinker) | F+A | 55.02 | **55.28** | 10.21 | 21.82 | 24.23 | **77.27** | 68.09 | 35.70 |

et al., 2024a) method designed for the molecular domain. By adopting the reverse process of the diffusion-based PC-Aware model, LinkerDPS samples a refined point cloud $(V, R)$ favored by the pretrained prior distribution of valid point clouds.

$$p(V, R \mid \tilde{G}, \mathcal{F}) = \frac{\overbrace{p(V, R \mid \mathcal{F})}^{\text{prior}} \cdot \overbrace{p(\tilde{G} \mid V, R, \mathcal{F})}^{\text{similarity}}}{p(\tilde{G} \mid \mathcal{F})} \tag{9}$$

As shown in equation 9, leveraging Bayes' rule, we decompose $p(V, R \mid \tilde{G}, \mathcal{F})$ into the prior distribution $p(V, R \mid \mathcal{F})$ and the likelihood $p(\tilde{G} \mid V, R, \mathcal{F})$, which respectively ensure the validity of $(V, R)$ and its similarity with $\tilde{G}$. Beyond the prior learned by the pretrained PC-Aware model, we design the likelihood of $\tilde{G} = (\tilde{V}, \tilde{E}, \tilde{R})$ as follows. We disregard the condition $\mathcal{F}$ in likelihood since $\tilde{G}$ always satisfies the condition. Additionally, we assume mutual independence of $\tilde{V}, \tilde{E}, \tilde{R}$ given $V$ and $R$.

**Likelihood of Atom.** To ensure consistency between the surrogate and generated molecules, we enforce that the atoms of both molecules remain the same. Formally, the likelihood of $\tilde{V}$ is defined as

$$\mathbb{P}_{\tilde{V}|V}(\tilde{V} \mid V) := \mathbb{I}(\tilde{V} = V) \tag{10}$$

where $\mathbb{I}$ denotes the indicator function.

**Likelihood of Bond and Conformation** To account for the cross-domain nature of bond likelihood given a point cloud, we introduce a molecular energy-inspired function $U$, defined as

$$U(E^*, R^*) := \sum_{1 \le i,j \le N} \mathbb{1}_{E_{i,j}^* \neq \mathbf{0}} \|R_i^* - R_j^*\|, \tag{11}$$

which quantifies the energy of a bond-conformation system forming a molecule. We define the cross-domain similarity of $\tilde{E}$ with a given $R$ and its likelihood as the probability that a molecule encompassing $R$ adopts the bond $\tilde{E}$. Utilizing the Boltzmann distribution, this quantity is expressed as being proportional to $\exp(-U(\tilde{E}, R))$. Similarly, to model the likelihood of $\tilde{R}$, we employ a Gaussian kernel $\kappa$ to quantify the similarity between $\tilde{R}$ and $R$ as $\kappa(\tilde{R}, R) = e^{-\|\tilde{R}-R\|^2}$, and we define the likelihood of $\tilde{R}$ to be proportional to the quantity. Combining these components, we formulate the joint likelihood of $(\tilde{E}, \tilde{R})$ as

$$p(\tilde{E}, \tilde{R} \mid R) := \frac{1}{Z}\kappa(\tilde{R}, R)e^{-U(\tilde{E},R)}, \tag{12}$$

where $Z$ is the normalization constant, assumed to be independent of $R$.

Accordingly, the likelihood of $\tilde{G} = (\tilde{V}, \tilde{E}, \tilde{R})$ in equation 9 is defined as

$$p(\tilde{G} \mid V, R, \mathcal{F}) = \mathbb{I}(\tilde{V} = V) \cdot \frac{1}{Z}\kappa(\tilde{R}, R)e^{-U(\tilde{E},R)} \tag{13}$$

**LinkerDPS Formulation** By equation 13, sampling refined point cloud $(V, R)$ via equation 9 reduces to sampling $R$ while fixing $V = \tilde{V}$, formulated as equation 14 where $\tilde{V} = V$ gives $p(\tilde{G} \mid R, V, \mathcal{F}) = p(\tilde{E}, \tilde{R} \mid R)$.

$$p(R \mid \tilde{G}, V, \mathcal{F}) \stackrel{\text{Bayes}}{\propto} p(R \mid V, \mathcal{F}) \cdot \underbrace{p(\tilde{G} \mid R, V, \mathcal{F})}_{=p(\tilde{E}, \tilde{R}|R)=equation\ 12}, \tag{14}$$

where $\mathbb{I}(\tilde{V} = V) = 1$ gives $p(\tilde{G} \mid R, V, \mathcal{F}) = p(\tilde{E}, \tilde{R} \mid R)$ by *equation* 13.

To sample $R$ from equation 14, we derive the reverse SDE in equation 15 and solve it using the pretrained score network of the PC-Aware model.

$$
\begin{aligned}
dr_t &= \left[ f(r_t, t) - g^2(t) \nabla_{r_t} \log p_t(r_t \mid \tilde{G}, V, \mathcal{F}) \right] dt + g(t) d\bar{W}_t \\
&\overset{\text{Bayes}}{=} \left[ f(r_t, t) - g^2(t) (\underbrace{\nabla_{r_t} \log p_t(r_t \mid V, \mathcal{F})}_{(a)} + \underbrace{\nabla_{r_t} \log p_t(\tilde{G} \mid r_t, V, \mathcal{F})}_{(b)}) \right] dt + g(t) d\bar{W}_t
\end{aligned}
\tag{15}
$$

In equation 15, $f(r_t, t) = -\frac{1}{2} \beta_t r_t$ and $g(t) = \sqrt{\beta_t}$ align with the reverse SDE of diffusion-based PC-Aware models in Equation (7) and $r_0 = R$. Thus, two $r_t$ in equation 15 and Equation (7) are equivalent.

We have (a) and (b) in Equation (15), not trivial to compute. To calculate (a), we apply inpainting trick (Lugmayr et al., 2022) on pretrained score network of PC-Aware model $s_{\theta^\star}(t, v_t, r_t \mid \mathcal{F}) \approx \nabla_{v_t, r_t} \log p_t(v_t, r_t \mid \mathcal{F})$ in Section 2.5 to implement score network additionally conditioned on $V$ without additional training:

$$
s_{\theta^\star}(t, r_t \mid V, \mathcal{F}) \approx \nabla_{r_t} \log p_t(r_t \mid V, \mathcal{F})
\tag{16}
$$

The detailed operation of this estimator is described in Appendix G.5. As for (b), since $p_t(\tilde{G} \mid r_t, V, \mathcal{F}) = \mathbb{E}_{r_0 \mid r_t, V, \mathcal{F}} \left[ p_t(\tilde{G} \mid r_0, V, \mathcal{F}) \right] = \mathbb{E}_{r_0 \mid r_t, V, \mathcal{F}} \left[ p_t(\tilde{E}, \tilde{R} \mid r_0) \right]$ as in Equation (14), we adapt the DPS approximation(Chung et al., 2024a) for our likelihood in equation 12, obtaining:

$$
\mathbb{E}_{r_0 \mid r_t, V, \mathcal{F}}[p_t(\tilde{E}, \tilde{R} \mid r_0)] \approx p(\tilde{E}, \tilde{R} \mid \hat{r}),
\tag{17}
$$

where $\hat{r} = \mathbb{E}_{r_0 \mid r_t, V, \mathcal{F}} [r_0]$ is tractable. Now, (b) is computable through simple additional steps. We describe the adaptation of the DPS approximation in Appendix G.3 and detailed computation of equation 15 using equation 16 and equation 17 in Appendix G.4.

Finally, we apply **ancestral sampling** (Ho et al., 2020a) to sample from the reverse SDE. The sampled conformation is concatenated with $V$ to construct the point cloud, which is subsequently converted into a complete molecular structure using the post-hoc bond predictor $\mathcal{E}$, following the PC-Aware model. The full algorithm for LinkerDPS is provided in Appendix G.6.

# 4 EXPERIMENTS

## 4.1 EXPERIMENT SETUP

We evaluate linker-generation algorithms on 400 fragment–linker pairs from ZINC-250K (Gómez-Bombarelli et al., 2018) and on GEOM-DRUG (Axelrod & Gomez-Bombarelli, 2021). Due to space limitations, GEOM-DRUG results are reported in Appx. H. Further, we run 50 times sampling for each fragment throughout the experiments.To evaluate the validity of generated samples, we use the metric **Validity**, based on the extended definition that considers the coherence between bonding topology and 3D conformation alongside traditional topological rules, as described in Appendix C. To score the diversity of samples, we use the followings to capture diversity in multiple perspective: **Uniqueness**, **Novelty**, **V+U**, **V+N**, **V+HD**, **V+FG**, and **V+BM**. Uniqueness and Novelty accounts for all samples satisfying topological valence rule, but the others measures the diversity of valid molecules by counting only the valid samples. More details of evaluation metrics in Appendix D.3. Furthermore, we present the results of additional experiments on ablation study and descriptor optimization, in Appendix L and Appendix I.

For baseline comparisons, we use all pretrained PC-Free and PC-Aware models trained on the ZINC dataset: FFLOM(Jin et al., 2023) and DeLinker(Imrie et al., 2020) for PC-Free models, and 3DLinker(Huang et al., 2022) and DiffLinker(Igashov et al., 2024) for PC-Aware models. We evaluate two implementations of HybridLinker, HybridLinker(FFLOM) and HybridLinker(DeLinker), both utilizing DiffLinker as the diffusion-based PC-Aware model while incorporating FFLOM and DeLinker as their respective PC-Free models. For all approaches, we use ETKDGv3(Riniker & Landrum, 2015) algorithm provided by RDkit(Landrum, 2022) as conformation predictor $\mathcal{R}$, and Obabel(O'Boyle et al., 2011) as post-hoc bond predictor $\mathcal{E}$. For further implementation details, see Appendix D.2.

Table 3: Comparison on Drug-Likeness Optimization of Reference Molecule. We report the percentage of input fragments in ZINC test data for which the model discovers the molecules surpassing the reference in drug-likeness metrics (lower QED, higher SA, higher PLogP, and their combinations).

| Model | QED (%) | SA (%) | PLogP (%) | Q&S (%) | S&P (%) | Q&P (%) | Q&S&P (%) |
|---|---|---|---|---|---|---|---|
| DiffLinker | 64.7 | 46.2 | 77.5 | 29.5 | 36.0 | 38.7 | 17.0 |
| FFLOM | 60.7 | 42.7 | 83.5 | 21.5 | 35.5 | 38.7 | 11.7 |
| HybridLinker (FFLOM) | 74.0 (+21.8%) | 51.0 (+19.3%) | 85.7 (+2.6%) | 30.5 (+41.8%) | 41.7 (+17.6%) | 48.5 (+25.1%) | 17.7 (+51.0%) |
| DeLinker | 74.2 | 59.7 | 83.2 | 41.5 | 49.2 | 49.7 | 24.7 |
| HybridLinker (DeLinker) | **76.7 (+3.3%)** | **61.7 (+3.3%)** | **84.7 (+1.8%)** | **43.2 (+4.2%)** | **52.2 (+6.0%)** | **54.0 (+8.5%)** | **27.0 (+9.0%)** |

## 4.2 RESULTS

**Advanced Diversity of Valid Molecules.** We run quantitative comparison of each linker generation algorithm in Table 2, demonstrating that HybridLinker successfully balance both high validity and diversity while all the baselines fail. The high scores in all diversity counting only valid moleules (V+U, V+N, V+HD, V+FG, V+BM), indicate that its diversity is driven by meaningful substructural variations. Notably, the high V+N score highlights HybridLinker's ability to discover non-trivial linkers for given fragments. When using DeLinker as the surrogate generator, HybridLinker achieves the highest validity, while utilizing surrogates from FFLOM boosts diversity while maintaining competitive validity. Importantly, HybridLinker also maintains high Uniqueness and Novelty scores, showing that its superior diversity of valid molecules stems from a balanced enhancement of both validity and diversity.

**Application: Drug-Likeness Optimization.** Drug-likeness optimization, which aims to identify molecules with better drug-likeness than reference design is a direct application benefitted by sample diversity. In Table 3, we demonstrate that the high diversity of HybridLinker enhances drug-likeness optimization results. In this experiment, we evaluate each linker generation algorithm regard to the success rate of drug-likeness optimization over the fragments-reference drug pairs in ZINC test dataset. To clarify, for each pair, the algorithm "success" in optimization if the generated samples contain molecule with drug-likeness score better than the reference molecule. We utilize three widely recognized drug-likeness scores—QED, SA, and PLogP—where higher values for QED and PLogP, and lower values for SA indicate better drug-likeness, and their combinations for stricter evaluation. The result highlights HybridLinker's strong performance, specifically, HybridLinker (DeLinker) outperfoms baselines in all scores while HybridLinker (FFLOM) significantly enhanced the performance of FFLOM.

## 4.3 DISCUSSION

**HybridLinker as a Foundational Model** As shown in Table 2, HybridLinker excels in linker generation by simultaneously achieving high diversity and validity—two fundamental pillars of drug discovery. Beyond its strong performance in core metrics, it also demonstrates exceptional results in application-driven tasks, as highlighted in Table 3. Moreover, as a training-free framework that seamlessly integrates pretrained PC-Free and PC-Aware models, HybridLinker remains highly adaptable to future advancements in these models, further enhancing its capabilities. These strengths establish HybridLinker as a foundational framework for linker generation, offering both versatility and long-term scalability in molecular design.

In the Appendix, we include: LinkerDPS–DPS distinctions (Appendix F); hyperparameter sensitivity (Appendix M); diversity–validity trade-offs and surrogate-quality effects (Appendices K and O); and inference time/cost (Appendix N).

## 5 CONCLUSION

We introduce HybridLinker, a training-free framework that integrates pretrained PC-Free and PC-Aware models to balance diversity and validity without additional training. At its core, LinkerDPS enables cross-domain guidance from molecular topology to point clouds, decomposing molecular sampling into sequential subtasks. Our results validate effectiveness of LinkerDPS as well as HybridLinker, and we anticipate its application in scalable large-molecule generation by breaking complex problems into manageable steps.

## 6 LLM USAGE DISCLOSURE

We used a large language model solely for light copyediting (spelling, grammar, and academic tone). The LLM did not contribute to research ideation, experimental design, implementation, or analysis. All scientific content and claims were written by the authors, who take full responsibility for the manuscript.

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

# A  NOTATION TABLE

Table 4: Notation table for molecular graph generation and related concepts.

| Symbol | Description |
|---|---|
| $G$ | A complete 3D molecular graph $G = (V, E, R) = (\mathcal{T}, R)$, where the number of atoms, $N$, is usually fixed to a reference value $N_{\text{ref}}$. |
| $V \in \{0, 1\}^{N \times A}$ | A list of atoms in $G$. $A$ is the number of atom types, and $N$ is the number of atoms. |
| $E \in \{0, 1\}^{N \times N \times B}$ | An adjacency tensor representing chemical bonds in $G$. |
| $R \in \mathbb{R}^{N \times 3}$ | A 3D coordinate matrix specifying the coordinates of each atom. |
| $\mathcal{T}$ | The bonding topology (2D representation) $\mathcal{T} = (V, E)$ which omits spatial information. |
| PC | The point cloud representation $(V, R)$, which omits bond information. |
| $\mathbb{P}_G = \mathbb{P}_{\mathcal{T}, R}$ | Distribution of valid 3D molecular graphs, or the joint distribution over bonding topology and 3D conformation. |
| $\mathcal{E}(\mathcal{F}, V, R)$ | Post-hoc bond predictor conditioned on the fragment condition $\mathcal{F}$. |
| $\mathcal{E}(\emptyset, V, R)$ | Unconditioned post-hoc bond predictor. |
| $\mathcal{R}(\mathcal{T}, R_1, R_2) \in \mathbb{R}^{N \times 3}$ | Off-the-shelf conformation generator given the topology $\mathcal{T}$ and fragment conformations $R_1, R_2$. |
| $\mathcal{T}_{\text{cond}}, R_{\text{cond}}$ | Given fragments $G_1 = (\mathcal{T}_1, R_1)$ and $G_2 = (\mathcal{T}_2, R_2)$:
$\quad \mathcal{T}_{\text{cond}} = \{G = (\mathcal{T}, R) : \mathcal{T}_1 \cup \mathcal{T}_2 \subset \mathcal{T}\}$,
$\quad R_{\text{cond}} = \{G = (\mathcal{T}, R) : R_1 \cup R_2 \subset R\}$. |
| $\mathcal{F}$ | Fragment condition: $\mathcal{F} = \mathcal{T}_{\text{cond}} \cap R_{\text{cond}} = \{G = (\mathcal{T}, R) : \mathcal{T}_1 \cup \mathcal{T}_2 \subset \mathcal{T},\ R_1 \cup R_2 \subset R\}$. |
| $\mathbb{P}_G(G \mid \mathcal{F})$ | $\mathbb{P}_{\mathcal{T}}(\mathcal{T} \mid \mathcal{F}) \cdot \mathbb{P}_{R \mid \mathcal{T}}(R \mid \mathcal{T}, \mathcal{F})$: true distribution used in PC-aware linker generation with spatial information. |
| $\mathbb{P}_{\text{2D}}(G \mid \mathcal{F})$ | $\mathbb{P}_{\mathcal{T}}(\mathcal{T} \mid \mathcal{T}_{\text{cond}}) \cdot \mathbb{P}_{R \mid \mathcal{T}}(R \mid \mathcal{T}, \mathcal{F})$: surrogate distribution used in PC-free linker generation without spatial information. |
| $\mathcal{U}$ | Molecular potential energy. |
| $s_{\theta^*}$ | Score estimator with pretrained parameter $\theta^*$. |
| $\mathbf{T}$ | A transformation in SE(3). |

## B  DETAILED DESCRIPTIONS FOR MOLECULAR REPRESENTATIONS

A molecule is commonly represented as a **molecular graph**, where nodes correspond to atoms, and edges represent covalent bonds(Huang et al., 2024). We formalize this below.

**Definition B.1** (3D Molecular Graph). A **3D molecular graph** of $N$ atoms is defined as a triplet $G = (V, E, R)$, where:

- $V \in \{0, 1\}^{N \times A}$ is the one-hot encoded list of **atoms** in the molecule where $A$ is the number of atom types.

- $E \in \{0, 1\}^{N \times N \times B}$ is the set of **chemical bonds**, represented as an adjacency tensor where $B$ is the number of bond types.

- $R \in \mathbb{R}^{N \times 3}$ denotes the **3D conformation**, specifying the spatial coordinates of each atom.

We also introduce two key sub-representations:

- The **bonding topology**, denoted as $\mathcal{T} = (V, E)$, which captures the connectivity of atoms without considering spatial information.

- The **point cloud representation**, given by $(V, R)$, which encodes atom types and 3D positions but omits bond information.

Thus, a 3D molecular graph can be equivalently expressed as $G = (\mathcal{T}, R)$, combining both connectivity and geometry. We write $\mathbb{P}_G = \mathbb{P}_{\mathcal{T}, R}$ to denote the distribution (dataset) of valid 3D molecular graphs which can be seen as the joint distribution over the bonding topology and 3D conformation.

**Point Cloud Representation for 3D Molecule Diffusion Models**  Recent molecular diffusion models(Hoogeboom et al., 2022; Igashov et al., 2024; Corso et al., 2023b; Schneuing et al., 2024; Corso et al., 2023a) represent molecules as point clouds $(V, R)$, disregarding explicit chemical bonds $(E)$ during the denoising steps. In these methods, the discrete point cloud representation is embedded in a continuous space $\mathbb{R}^d$, where $d = N(A + 3)$ accounts for both atom types (one-hot encoded) and spatial coordinates. Diffusion denoising steps operate in $\mathbb{R}^d$, gradually refining the continuous representation. After sampling, the discrete atom representation is recovered using an `argmax` operation on $V$. Chemical bonds are inferred via a post-hoc bond predictor $\mathcal{E}$ as $E = \mathcal{E}(\mathcal{F}, V, R)$ or $\mathcal{E}(\emptyset, V, R)$ depending on fragment conditions $\mathcal{F}$, producing $G = (V, E, R)$.

## C  EXTENDED DEFINITION OF VALIDITY

In this work, we extend the previous definition of molecule validity, focusing on each topology representation, to reflect its point cloud representations, which is a crucial consideration to make the the validity indicates its stability in 3D space where they serve a drug. We accounts for not only the valence rule in molecular topology but also the coherence of molecular point cloud with the topology. Molecular potential energy $\mathcal{U}$ is adopted to score the pair of topology and point cloud. We determine the validity of given molecule based on its strain(conformation) energy, the gap of its energy the its optimal energy attained in its most stable conformation without external constraints as fragments condition.

To clarify, higher conformation energy means the molecule is unstable, also can be interpreted by Boltzmann distribution. **In our setup of linker generation, the output molecule can be unstable (high conformation energy) if the generated linker is not appropriate for the given fragments' 3D pose condition, i.e., some bonds should be stretched to connect fragments.** We choose the widely used cut-off 25(kcal / mol)(Sitzmann et al., 2012; Kirchmair et al., 2006; Peach et al., 2017) of practically allowable conformation energy as our threshold $\tau^{\text{val}}$, and label the molecule valid if its conformation energy is lower than the $\tau^{\text{val}}$. We use RDKit(Landrum, 2022) to compute $\mathcal{U}$ and molecule's optimal conformation.

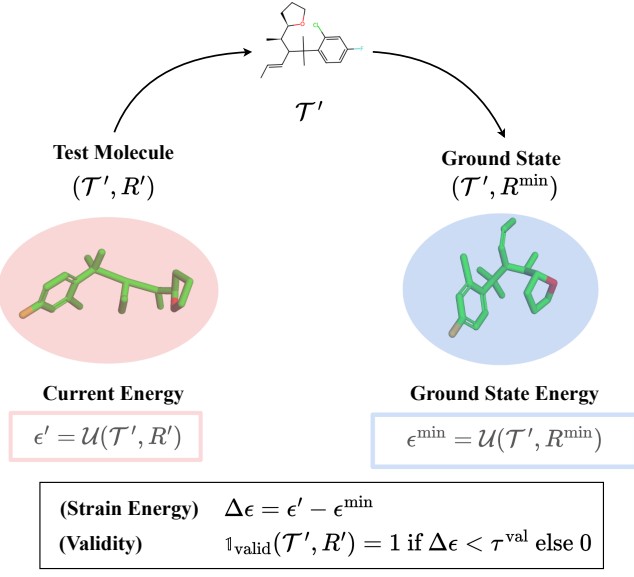

Figure 5: Extended definition of molecular validity, accounting for its point cloud feature as well as topological valence rule via utilizing strain energy.

## D    EXPERIMETAL DETAILS

### D.1    DATASET CONSTRUCTION

Following DeLinker(Imrie et al., 2020), we select a subset of 250,000 molecules from the ZINC dataset. The energetically stable conformation of each molecule is determined using MMFF force field optimization(Halgren, 1996) implemented in RDKit(Landrum, 2022). As in prior work(Hussain & Rea, 2010), molecular fragmentation is performed by applying double bond cuts on acyclic single bonds that are not part of functional groups. Following the preprocessing steps in DeLinker(Imrie et al., 2020), we construct 418,797 fragment-linker pairs and further select 400 pairs as our test set, aligning with the test set used in DeLinker(Imrie et al., 2020). In this study, we exclusively use the test set to evaluate HybridLinker, as our method does not require additional training but instead operates in a training-free manner using pretrained models.

### D.2    IMPLMENTATION OF BASELINES AND HYBRIDLINKER

We compare HybridLinker with recent baselines for linker generation, falls into either PC-Free models or PC-Aware models. FFLOM(Jin et al., 2023) and DeLinker(Imrie et al., 2020) are included to represents PC-Free models. For the PC-Free models, we adopt the strong and easy-to-use conformation generation algorithm, ETKDGv3(Riniker & Landrum, 2015), to predict conformation of molecule topology they generate. For PC-Aware models, we include 3DLinker(Huang et al., 2022) and DiffLinker(Igashov et al., 2024) as baseline models.

HybridLinker is implemented utilizing both pretrained PC-Free models and PC-Aware models, respectively. As far we know, DiffLinker is the only strong PC-Aware baseline, and we adopt it in HybridLinker's second stage. As for the PC-Free model in the first step, we utilze FFLOM and DeLinker, as the two are only strong PC-Free baselines. In the experiment, we refer the PC-Free model in HybridLinker's first stage as surrogate generator. Note that we do not conduct additional training of baselines. Further, we found that if surrogate molecule $\tilde{G}$ from the first stage is already valid, skipping second stage to sample $G$ but directly sample $G = \tilde{G}$ is effective to achieve high performance. We follow this scheme in our experiments.

Further, we use ETKDGv3(Riniker & Landrum, 2015) algorithm provided by RDkit(Landrum, 2022) as the off-the-shelf conformation predictor $\mathcal{R}$, and Obabel(O'Boyle et al., 2011) as post-hoc bond predictor $\mathcal{E}$.

### D.3    EVALUATION METRICS

In Section 4.2, we first evaluate the samples generated by each algorithm based on their validity and diversity. Notably, we extend the definition of **Validity** from prior works to account for both valence rules in bonding topology and energetic stability in the 3D graph. To evaluate sample diversity, we incorporate standard metrics such as **Uniqueness** and **Novelty**. However, unlike prior works that define Novelty as the fraction of novel molecules, we compute it as the fraction of molecules that are both unique and novel. Additionally, we introduce three more diversity metrics introduced in (Hu et al., 2024)—**HamDiv, FG, BM**—to capture diversity from different perspectives, specifically, molecule fingerprint, number of unique functional group, and number of unique molecular scaffolds, respectively. We follow the implementation in the repository (https://github.com/HXYfighter/HamDiv) to caculate them. We also heavily focus on calculating each diversity score counting only the valid molecules (diversity metrics for valid molecules).In the main paper, we denote them as V+U (Uniqueness), V+N (Novelty), V+HD (HamDiv), V+FG (FG), and V+BM (BM). For instance, the calculation of V+U for a given fragment pair differs from that of Uniqueness as follows:

$$\text{Uniqueness} = \frac{|\mathcal{S}|}{N},$$
$$\text{V} + \text{U} = \frac{|\{\mathcal{T} \mid \mathcal{T} \in \mathcal{S}, \ \mathcal{T} \text{ is valid}\}|}{N}, \tag{18}$$

where $\mathcal{S} = \{\mathcal{T}_i\}_{i=1}^{N}$ represents the set of topologies for the $N$ generated molecules corresponding to the fragment pair. The other diversity metrics for valid molecules are computed in a similar manner.

All metrics above are first calculated for 50 samples for each fragments, and averaged over the test fragments.

We also evaluate the drug-likeness and chemical properties of the generated samples. In Section 4.2, we compare the drug-likeness of samples from each algorithm. For drug-likeness scoring, we use **QED**(Bickerton et al., 2012), **SA**(Ertl & Schuffenhauer, 2009), and **PLogP**(You et al., 2018), which measure general drug quality, synthetic accessibility, and the octanol-water partition coefficient penalized by the synthetic accessibility score, respectively.

## E  ALGORITHMIC COMPARISON ON PC-AWARE MODEL, PC-FREE MODEL, AND HYBRIDLINKER

Here, we illustrate the two-step generation pipelin of HybridLinker (Figure 6) and algorithmic distinction of HybridLinker compared to PC-Aware and PC-Free models(Algorithm 1, Algorithm 2, Algorithm 3). While PC-Aware and PC-Free models perform either PC-Aware or PC-Free inference, specializing in validity or diversity respectively, HybridLinker integrates both inference types in two-step pipeline, effectively leveraging the strengths of both approaches.

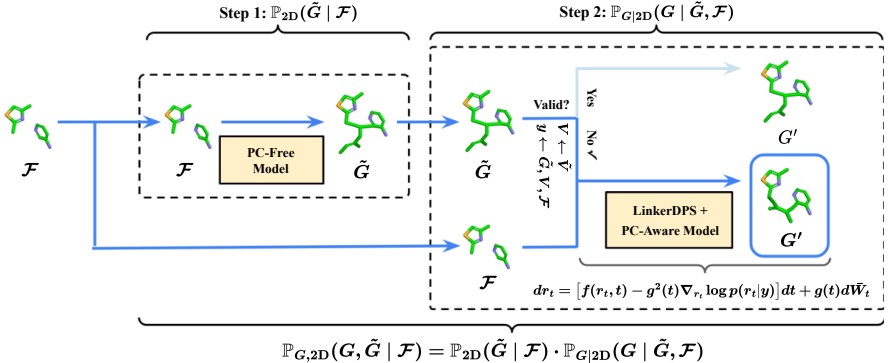

$$\mathbb{P}_{G,\text{2D}}(G, \tilde{G} \mid \mathcal{F}) = \mathbb{P}_{\text{2D}}(\tilde{G} \mid \mathcal{F}) \cdot \mathbb{P}_{G|\text{2D}}(G \mid \tilde{G}, \mathcal{F})$$

Figure 6: Two-step generation pipeline of HybridLinker. (Step 1.) diverse molecular samples are generated using pretrained PC-Free models. (Step 2.) posterior sampling is performed for each molecule from the previous step. If a molecule is invalid, LinkerDPS is applied to a diffusion-based PC-Aware model to refine it into a valid structure while maintaining similarity to the guiding molecule. Valid molecules from the first step are directly used as the final output.

---

**Algorithm 1** PC-Aware Linker Generation Model (Diffusion-based)

---

1: **INPUT** fragments $G_1 = (\mathcal{T}, R_1) = (V_1, E_1, R_1)$, $G_2 = (\mathcal{T}_2, R_2) = (V_2, E_2, R_2)$, PC-Aware condition $\mathcal{F} = G_1 \cap G_2$, pretrained score estimator $s_{\theta^*}$, bond predictor $\mathcal{E}$, variance level $\{\bar{\beta}_t = 1 - \exp(-\int_0^t \beta_u du)\}_{t=1}^T$

2: $x_T \sim \mathcal{N}(0, I)$         ◁ run reverse process for atom generation

3: **for** $t = T, \ldots, 1$ **do**

4:    $\hat{s} \leftarrow s_{\theta^*}(t, x_t \mid \mathcal{F})$         ◁ estimate score $\nabla_{x_t} \log p_t(x_t)$

5:    $z \sim \mathcal{N}(0, I)$

6:    $\hat{x} \leftarrow \frac{1}{\sqrt{1-\bar{\beta}_t}}(x_t + \bar{\beta}_t \hat{s})$         ◁ compute $\hat{x} \approx \mathbb{E}_{x_0|x_t}[x_0]$

7:    $x_{t-1} \leftarrow \frac{\sqrt{\alpha_t}(1-\bar{\alpha}_{t-1})}{1-\bar{\alpha}_t}x_t + \frac{\sqrt{\bar{\alpha}_{t-1}}\beta_t}{1-\bar{\alpha}_t}\hat{x} + \sqrt{\bar{\beta}_t}z$     ◁ update $x_{t-1}$

8: **end for**

9: $(v', R') \leftarrow x_0$

10: $V' \xleftarrow{\text{argmax}} v'$

11: $E' \leftarrow \mathcal{E}(V', R')$

12: $G' \leftarrow (V', E', R')$

13: **return** $G'$

---

**Algorithm 2** PC-Free Linker Generation Model

---

1: **INPUT** fragments $G_1 = (\mathcal{T}, R_1) = (V_1, E_1, R_1)$, $G_2 = (\mathcal{T}_2, R_2) = (V_2, E_2, R_2)$, PC-Free condition $\mathcal{T}_{\text{cond}} = \mathcal{T}_1 \cap \mathcal{T}_2$, PC-Aware condition $\mathcal{F} = G_1 \cap G_2$, parameterized function for PC-Free model $f_{\theta F}$, conformation predictor $\mathcal{R}$

2: $\mathcal{T}' \leftarrow f_{\theta F}(\mathcal{T}_{\text{cond}})$     ◁ sample topology from topology condition: $\mathcal{T}' \sim \mathbb{P}_{\mathcal{T}|\mathcal{T}_{\text{cond}}}$

3: $R' \leftarrow \mathcal{R}(\mathcal{T}', \mathcal{F})$     ◁ conformation prediction $R' \sim \mathbb{P}_{R|\mathcal{T}}(R', \mathcal{T}')$

4: $G' \leftarrow (\mathcal{T}' \mid R', \mathcal{F})$

5: **return** $G'$

---

---

**Algorithm 3** HybridLinker

---

1: **INPUT** fragments $G_1 = (\mathcal{T}, R_1) = (V_1, E_1, R_1)$, $G_2 = (\mathcal{T}_2, R_2) = (V_2, E_2, R_2)$, PC-Free condition $\mathcal{T}_{\text{cond}} = \mathcal{T}_1 \cap \mathcal{T}_2$, PC-Aware condition $\mathcal{F} = G_1 \cap G_2$, pretrained score estimator for diffusion-based PC-Aware model $s_{\theta*}$, parameterized function for PC-Free model $f_{\theta F}$, conformation predictor $\mathcal{R}$, bond predictor $\mathcal{E}$

2: $\tilde{\mathcal{T}} \leftarrow f_{\theta F}(\mathcal{T}_{\text{cond}})$

3: $\tilde{R} \leftarrow \mathcal{R}(\mathcal{T}', \mathcal{F})$

4: $\tilde{G} \leftarrow (\tilde{\mathcal{T}}, \tilde{R})$                  ◁ 1st Stage (PC-Free Generation): $\tilde{G} \sim \mathbb{P}_{\text{2D}}(\tilde{G} \mid \mathcal{F})$

5: **if** $\tilde{G}$ is invalid **then**

6:      $(V', R') \leftarrow$ **LinkerDPS**$(s_{\theta*}, \mathcal{F}, \tilde{G})$        ◁ For invalid $\tilde{G}$, run LinkerDPS with PC-Aware model to enhance validity

7:      $E \leftarrow \mathcal{E}(V', R')$

8:      $G' \leftarrow (V', E', R')$

9: **else**

10:     $G' \leftarrow \tilde{G}$                            ◁ For valid $\tilde{G}$, set $G = \tilde{G}$

11: **end if**               ◁ 2nd Stage (PC-Aware Generation): $G' \sim \mathbb{P}_{G|\text{2D}}(G' \mid \tilde{G}, \mathcal{F})$

12: **return** $G'$

---

# F  KEY DISTINCTIONS: LINKERDPS VS. DPS

Here, we highlight the difference between DPS (Chung et al., 2024a) and LinkerDPS, to show that LinkerDPS is not the straghtforward application of DPS.

Table 5: **Feature comparison.** Yes = supported; No = not supported.

| Feature / Approach | DPS (Chung et al.) | LINKERDPS (Ours) |
|---|---|---|
| Training-free conditional generation | Yes | Yes |
| Tractable likelihood | Yes | Yes |
| Extends beyond DPS theorem | No | Yes |
| Cross-domain guidance | No | Yes |
| Molecular graph conditioning | No | Yes |

*Notes.* For "Tractable likelihood", DPS assumes additive noise within a *single* (same) domain; ours introduces a cross-domain energy-based likelihood tailored to molecular graphs and 3D coordinates. "Extends beyond DPS theorem" indicates that our guarantees require Theorem F.4 (extending Theorem 1 of Chung et al.).

**LinkerDPS is not the straightforward application of DPS.**  DPS is derived for continuous domains with additive noise (e.g., Gaussian/Poisson) and a measurement model $y = \mathcal{H}(x) + \varepsilon$, which yields an analytic likelihood $p(y \mid x)$ and enables posterior sampling for recovering $x$ from $y$. In our setting: (i) the input is a surrogate molecule from a PC-Free generator; (ii) the target is a true molecular structure; (iii) no additive noise model exists; and (iv) data live in a hybrid discrete–continuous space (graphs + 3D coordinates). This mismatch makes the original DPS formulation inapplicable as-is.

**Our contribution: a cross-domain likelihood for molecules.**  We construct an energy-based likelihood that aligns fragment-conditioned molecules across modalities (Eqs. 11–13):

$$\log p(y \mid x) \ \propto \ -E_{\text{atom}}(x, y) - \omega_b\, E_{\text{bond}}(x, y) - \omega_c\, E_{\text{coord}}(x, y),$$

where discrete atom types are enforced via inpainting in $E_{\text{atom}}$, bond topology is softly guided by a cross-domain energy $E_{\text{bond}}$, and 3D geometry is matched via a Gaussian kernel in $E_{\text{coord}}$. Crucially, we prove that the DPS approximation (including the upper bound on the Jensen gap) continues to hold under this model; see Theorem F.4 in the supplement, which extends Chung et al.'s Theorem 1 to the cross-domain, graph+geometry case.

**Motivation for a DPS-like design.**  Rather than training a new model (and re-incurring validity–diversity trade-offs), we cast linker generation as an inverse problem: recover a valid, pocket-aware molecule from a coarse PC-Free surrogate. The derivation leading to Eq. (8) shows that a DPS-style sampler is well-suited here—leveraging pretrained diffusion denoisers for training-free, effective inference. Unlike image settings, prior work has not applied DPS to molecular *graph* generation under fragment constraints; LinkerDPS bridges physics-inspired modeling with modern diffusion sampling in this new domain.

# G   DETAILS OF LINKERDPS

## G.1   OVERVIEW ON DIFFUSION POSTERIOR SAMPLING

Diffusion Posterior Sampling (DPS)(Chung et al., 2024a) is a novel method for addressing noisy inverse problems using diffusion models. Diffusion models, which are typically employed for generative tasks, works in reverse process of diffusion forward pass that progressively add noise to data.

**Formulation of Generative SDE.** Formally, diffusion process is governed by following SDE:

$$dx_t = f(x_t, t)dt + g(t)dW_t, \quad t \in [0, T] \tag{19}$$

where $f(x_t, t) = -\frac{1}{2}\beta_t x_t$ is the drift coefficient and $g(t) = \sqrt{\beta_t}$ is the diffusion coefficient for the noise scheduler $\{\beta_t\}$. Further, $W_t$ is the standard Wiener process and $dt$ is an infinitesimal timestep.

To recover the data generating distribution, we can use the reverse SDE defined as

$$dx_t = \left[f(x_t, t) - g^2(t)\nabla_{x_t} \log p(x_t)\right] dt + g(t)d\bar{W}_t, \quad t \in [0, T] \tag{20}$$

where $\bar{W}$ is a reversed Wiener process flows backward from $t = 1$ to $t = 0$. Here, since the score $\nabla_{x_t} \log p(x_t)$ is intractable, the neural score network $s_{\theta*}$ is used for computation where

$$\theta^* = \arg \min_\theta \mathbb{E}_{t \sim u(\epsilon, 1), x_t \sim p(x_t | x_0), x_0 \sim p_{\text{data}}} \left[\|s_\theta(x_t, t) - \nabla_{x_t} \log p(x_t | x_0)\|^2\right] \tag{21}$$

for a very small positive constant $\epsilon \simeq 0$.

**Conditional Sampling via DPS.** Now, we sample data from the posterior $p(x|y)$ given noisy measurement $y$, where the likelihood of measurement $y$ given $x$ is formulated as $p(y \mid x) \propto \exp\left(-\frac{\|y - \mathcal{A}(x)\|^2}{\sigma}\right)$ for some $\sigma \in \mathbb{R}$. The authors (Chung et al., 2024a) also show the application in several other likelihoods.

Based on the Bayes' rule, the new score $\nabla_{x_t} \log p(x_t|y)$ is decomposed into

$$\nabla_{x_t} \log p(x_t|y) = \nabla_{x_t} \log p(x_t) + \nabla_{x_t} \log p(y|x_t). \tag{22}$$

The proposed DPS approximation (Theorem G.3 ) makes the intractable term $\nabla_{x_t} \log p(y|x_t)$ computable as

$$\nabla_{x_t} \log p(y|x_t) \approx \nabla_{x_t} \hat{x}(x_t) \nabla_{x_0} \log p(y|x_0 = \hat{x}) \tag{23}$$

where $\hat{x} = \mathbb{E}_{x_0|x_t}[x_0]$. Note that $\mathbb{E}_{x_0|x_t}[x_0]$ can be estimated using the trained score network $s_{\theta*}$ as $\frac{1}{\sqrt{1-\bar{\beta}_t}}\left(x_t + \bar{\beta}_t s_{\theta*}(x_t, t)\right)$ for $\bar{\beta}_t = 1 - \exp\left(-\int_0^t \beta_u du\right)$, since $x_t \sim \mathcal{N}(\sqrt{1 - \bar{\beta}_t}x_0, \bar{\beta}_t I)$ and $s_{\theta*}(x_t, t)$ estimates $(\sqrt{1 - \bar{\beta}_t}x_0 - x_t)/\bar{\beta}_t$. Now, equation 23 is computable using the backward-propagation on tractable computation of $\hat{x}$ and $\log p(y|x_0)$. Consequently, the computable SDE for conditional generation from $p(x \mid y)$ becomes

$$dx_t = \left[f(x_t, t) - g^2(t)(\nabla_{x_t} \log p(x_t) + \nabla_{x_t} \hat{x}_0(x_t) \nabla_{x_0} \log p(y|x_0 = \hat{x}))\right] dt + g(t)d\bar{W}_t, \quad t \in [0, T] \tag{24}$$

## G.2 DPS APPROXIMATION

Exploiting Definition equation G.1, (Chung et al., 2024a) proposed Theorem G.3 about DPS approximation.

**Definition G.1** (Jensen Gap (Gao et al., 2017))**.** Let $x$ be a random variable with distribution $p(x)$. For some function $f$ that may or may not be convex, the Jensen gap is defined as

$$\mathcal{J}(f, x \sim p(x)) = \mathbb{E}[f(x)] - f(\mathbb{E}[x]), \tag{25}$$

where the expectation is taken over $p(x)$.

**Definition G.2.** The general form of the forward model of an inverse problem can be stated as

$$y = \mathcal{A}(x_0) + \epsilon \tag{26}$$

where $y, \epsilon \in \mathbb{R}^n, x_0 \in \mathbb{R}^d$ and $\mathcal{A}(\cdot) : \mathbb{R}^d \mapsto \mathbb{R}^n$ is the forward measurement operator and $\epsilon$ is the measurement noise.

**Theorem G.3.** *For the measurement model defined in Definition equation G.2 with $\epsilon \sim \mathcal{N}(0, \sigma^2 I)$, we have*

$$p(y \mid x_t) \approx p(y \mid x_0 = \hat{x}) \tag{27}$$

*with $\hat{x} = \mathbb{E}_{x_0 \mid x_t}[x_0]$ where the approximation error can be quantified with the Jensen gap, which is upper bounded by*

$$\mathcal{J} \leq \frac{d}{\sqrt{2\pi\sigma^2}} e^{-1/2\sigma^2} \|\nabla_x \mathcal{A}(x)\| m_1, \tag{28}$$

*where $\|\nabla_x \mathcal{A}(x)\| := \max_x \|\nabla_x \mathcal{A}(x)\|$ and $m_1 := \int \|x_0 - \hat{x}\| p(x_0 \mid x_t) \, dx_0$.*

### G.3 ADAPTATION OF DPS APPROXIMATION

Inspired by Theorem G.3, we propose Theorem G.4 about LinkerDPS approximation exploiting Lemma equation G.5 and Proposition equation G.6. Theorem G.4 and the subsequent computation

$$\mathbb{E}_{r_0|r_t,V,\mathcal{F}}[r_0] = \frac{1}{\sqrt{1-\bar{\bar{\beta}}_t}}\left(r_t + \bar{\beta}_t \cdot s_{\theta^*}(t, r_t \mid V, \mathcal{F})\right), \tag{29}$$

which refers Appendix G.1 and equation 16, validate equation 17 and make (b) of equation 15 computable.

**Theorem G.4.** *(LinkerDPS Approximation) Suppose we have the cross domain measurement model,*

$$p(\tilde{E}, \tilde{R} \mid r) = \frac{1}{Z}e^{-\left[\phi_1(r|\tilde{R})/2\sigma_1^2 + \phi_2(r|\tilde{E})/\sigma_2\right]} \tag{30}$$

*where $\sigma_1, \sigma_2 \in \mathbb{R}$ and $\phi_1, \phi_2$ are defined as*

$$\phi_1(r \mid \tilde{R}) := \|\tilde{R} - r\|^2,$$
$$\phi_2(r \mid \tilde{E}) := \sum_{1 \le i,j \le N} \mathbb{1}_{\tilde{E}_{i,j} \neq \mathbf{0}} \|r_i - r_j\|. \tag{31}$$

*In addition, $\{r_t\}_{t\in[0,1]}$ is the diffusion process of conformation $r_0 = R$ following the forward SDE $dr_t = -\frac{1}{2}\beta_t r_t + \sqrt{\beta_t}dW$. Then, we have*

$$\mathbb{E}_{r_0|r_t,V,\mathcal{F}}[p(\tilde{E}, \tilde{R} \mid r_0)] \approx p(\tilde{E}, \tilde{R} \mid \hat{r}), \tag{32}$$

*where $\hat{r} = \mathbb{E}_{r_0|r_t,V,\mathcal{F}}[r_0]$ and the approximation error for equation 32 can be quantified with the Jensen gap $\mathcal{J}$, which is upper bounded by*

$$\mathcal{J} \le Lm_1, \tag{33}$$

*where $L = \frac{1}{Z} \cdot \left(\frac{e^{-1/2}}{\sigma_1} + \frac{N^{3/2}}{\sigma_2}\right)$, and $m_1 := \int \|r_0 - \hat{r}\| p(r_0|r_t, V, \mathcal{F}) \, dr_0$.*

*proof of Theorem G.4.*

For $f(r) := p(\tilde{E}, \tilde{R} \mid r)$, the approximation error of equation 32 becomes the Jensen gap $\mathcal{F}$ as

$$\mathcal{J} = |\mathbb{E}_{r_0|r_t,V,\mathcal{F}}[f(r_0)] - f(\mathbb{E}_{r_0|r_t,V,\mathcal{F}}[r_0])| \tag{34}$$

Let $\hat{r} = \mathbb{E}_{r_0|r_t,V,\mathcal{F}}[r_0]$ and leveraging Lemma equation G.5,

$$|f(r_0) - f(\hat{r})| \le L\|r_0 - \hat{r}\| \tag{35}$$

where $L = \frac{1}{Z} \cdot \left(\frac{e^{-1/2}}{\sigma_1} + \frac{N^{3/2}}{\sigma_2}\right)$. Then, according to Proposition equation G.6, $\mathcal{J}$ has upper bound as

$$\mathcal{J} \le Lm_1 \tag{36}$$

where $m_1 = \int \|r_0 - \hat{r}\| p(r_0|r_t, V, \mathcal{F}) \, dr_0$. $\square$

**Lemma G.5.** *Let $f(r) := p(\tilde{E}, \tilde{R} \mid r)$ in equation 30. There exists a constant $L$ such that $\forall r, r^* \in \mathbb{R}^{3N}$,*

$$|f(r) - f(r^*)| \le L\|r - r^*\|, \tag{37}$$

*where $L = \frac{1}{Z} \cdot \left(\frac{e^{-1/2}}{\sigma_1} + \frac{N^{3/2}}{\sigma_2}\right)$.*

*proof of Lemma equation G.5.*

$$
\begin{aligned}
\max_r \|\nabla_r f(r)\| &= \max_r \|f(r) \cdot \left(-\frac{1}{2\sigma_1^2} \cdot \nabla_r \phi_1(\tilde{R} \mid r) - \frac{1}{\sigma_2} \cdot \nabla_r \phi_2(\tilde{E} \mid r)\right)\| \\
&\le \max_r \|f(r) \cdot \left(-\frac{1}{2\sigma_1^2} \cdot \nabla_r \phi_1(\tilde{R} \mid r)\right)\| + \max_r \|f(r) \cdot \left(-\frac{1}{\sigma_2} \cdot \nabla_r \phi_2(\tilde{E} \mid r)\right)\| \\
&= \max_r \|f(r) \cdot \left(-\frac{r - \tilde{R}}{\sigma_1^2}\right)\| + \max_r \|f(r) \cdot \left(-\frac{1}{\sigma_2} \cdot \nabla_r \phi_2(\tilde{E} \mid r)\right)\| \\
&\le \frac{1}{\sigma_1} \cdot \max_r \left(f(r) \cdot \|\frac{r - \tilde{R}}{\sigma_1}\|\right) + \frac{1}{\sigma_2} \cdot \max_r \|f(r) \cdot \nabla_r \phi_2(\tilde{E} \mid r)\| \\
&\le \frac{1}{Z} \cdot \left(\frac{e^{-1/2}}{\sigma_1} + \frac{N^{3/2}}{\sigma_2}\right)
\end{aligned}
\tag{38}
$$

where the last two inequalities are from

$$
\begin{aligned}
\max_r \left(f(r) \cdot \|\frac{r - \tilde{R}}{\sigma_1}\|\right) &\le \max_r \left(\frac{e^{-\|r - \tilde{R}\|^2/(2\sigma_1^2)}}{Z} \cdot \|\frac{r - \tilde{R}}{\sigma_1}\|\right) \\
&= \max_{z \in \mathbb{R}^+} \frac{z \cdot e^{-z^2/2}}{Z} \\
&= \frac{e^{-1/2}}{Z}
\end{aligned}
\tag{39}
$$

and

$$
\begin{aligned}
\max_r \|f(r) \cdot \nabla_r \phi_2(\tilde{E} \mid r)\| &\le \frac{1}{Z} \cdot \max_r \|\nabla_r \phi_2(\tilde{E} \mid r)\| \\
&\le \frac{1}{Z} \cdot \sqrt{N \cdot (N)^2} \\
&\le \frac{N^{3/2}}{Z}
\end{aligned}
\tag{40}
$$

Then, we have

$$
\begin{aligned}
|f(r) - f(r^*)| &\le \max_r \|\nabla_r f(r)\| \cdot \|r - r^*\| \\
&\le \frac{1}{Z} \cdot \left(\frac{e^{-1/2}}{\sigma_1} + \frac{N^{3/2}}{\sigma_2}\right) \cdot \|r - r^*\|
\end{aligned}
\tag{41}
$$

$\square$

**Proposition G.6** (Jensen gap upper bound(Gao et al., 2017))**.** *Define the absolute cenetered moment as $m_p := \sqrt[p]{\mathbb{E}[|X - \mu|^p]}$, and the mean as $\mu = \mathbb{E}[X]$. Assume that for $\alpha > 0$, there exists a positive number $K$ such that for any $x \in \mathbb{R}^d$, $|f(x) - f(\mu)| \le K|x - \mu|^\alpha$. Then,*

$$
\begin{aligned}
|\mathbb{E}[f(X)] - f(\mathbb{E}[X])| &\le \int |f(X) - f(\mu)| dp(X) \\
&\le K \int |x - \mu|^\alpha dp(X) \le K m_\alpha^\alpha.
\end{aligned}
\tag{42}
$$

### G.4 DETAILED COMPUTATION OF LINKERDPS

We show that reverse SDE for LinkerDPS (Equation (15)) is tractable as follows. Substituting Equation (16) and Equation (17) respectively in (a) and (b) of Equation (15) give

$$
\begin{aligned}
dx_t \quad &= \Big[ f(r_t, t) - g^2(t)(s_{\theta^\star}(t, r_t \mid V, \mathcal{F}) + \nabla_{r_t} \log p(\tilde{E}, \tilde{R} \mid \hat{r})) \Big] dt + g(t) d\bar{W}_t \\
&\overset{\text{Chain Rule}}{=} \Big[ f(r_t, t) - g^2(t)(s_{\theta^\star}(t, r_t \mid V, \mathcal{F}) + \underbrace{\nabla_{r_t} \hat{r}}_{\texttt{autograd}} \cdot \underbrace{\nabla_{\hat{r}} \log p(\tilde{E}, \tilde{R} \mid \hat{r})}_{equation\ 44, equation\ 45} ) \Big] dt + g(t) d\bar{W}_t
\end{aligned}
\tag{43}
$$

where $\hat{r} = \mathbb{E}_{r_0 \mid r_t, V, \mathcal{F}} [r_0] = \frac{1}{\sqrt{1 - \bar{\beta}_t}} \big( r_t + \bar{\beta}_t \cdot s_{\theta^\star}(t, r_t \mid V, \mathcal{F}) \big)$ for $\bar{\beta}_t = 1 - \exp\Big( - \int_0^t \beta_u du \Big)$.

We clarify that $\nabla_{r_t} \hat{r} \cdot \nabla_{\hat{r}} \log p(\tilde{E}, \tilde{R} \mid \hat{r})$ is tractable. First, $\nabla_{r_t} \hat{r}$ is computed using `autograd` after the forward calculation. The latter $\nabla_{\hat{r}} \log p(\tilde{E}, \tilde{R} \mid \hat{r})$ is expressed as

$$
\nabla_{\hat{r}} \log p(\tilde{E}, \tilde{R} \mid \hat{r}) = -\nabla_{\hat{r}} \phi_1(\hat{r} \mid \tilde{R}) - \nabla_{\hat{r}} \phi_2(\hat{r} \mid \tilde{E}),
\tag{44}
$$

where $\phi_1(\hat{r} \mid \tilde{R}) := \|\tilde{R} - \hat{r}\|^2$ and $\phi_2(\hat{r} \mid \tilde{E}) := \sum_{1 \le i, j \le N} \mathbb{1}_{\tilde{E}_{i,j} \neq \mathbf{0}} \|\hat{r}_i - \hat{r}_j\|$.

Indeed, $\nabla_{\hat{r}} \phi_1(\hat{r} \mid \tilde{R})$ and $\nabla_{\hat{r}} \phi_2(\hat{r} \mid \tilde{E})$ are computed in straightforward as

$$
\begin{aligned}
\nabla_{\hat{r}_i} \phi_1(\hat{r} \mid \tilde{R}) &= 2(\hat{r}_i - \tilde{R}_i)^T \\
\nabla_{\hat{r}_i} \phi_2(\hat{r} \mid \tilde{E}) &= \sum_{1 \le j \le N} \frac{\mathbb{1}_{\tilde{E}_{i,j} \neq \mathbf{0}} (\hat{r}_i - \hat{r}_j)^T}{\|\hat{r}_i - \hat{r}_j\|},
\end{aligned}
\tag{45}
$$

which makes equation 44 tractable.

## G.5 CONDITIONAL SCORE ESTIMATOR

Leveraging pretrained score estimator $s_{\theta^*}$ of PC-Aware model (Igashov et al., 2024)

$$s_{\theta^*}(t, v_t, r_t, \mid \mathcal{F}) \approx \nabla_{v_t, r_t} \log p_t(v_t, r_t \mid \mathcal{F}), \tag{46}$$

for joint sampling of atom type $v_0 = V$ and conformation $r_0 = R$, we derive a conditional score estimator for sampling of conformation alone under the fixed atom type. Specifically, the conditional score estimator predicts the score of the conditional distribution $p_t(r_t \mid V, \mathcal{F})$.

To achieve this, we treat the conditional sampling as the inpainting problem and adapt the methods of (Lugmayr et al., 2022) to 3D molecules. To clarify, we formulate the problem as the sampling of molecule $X = (V, R)$, where we know the regions for the atom types $V$ and the remaining regions for conformation $R$ should be sampled via backward diffusion process $\{x_t = (v_t, r_t)\}_{0 \leq t \leq T}$. Since every reverse step from $x_t$ to $x_{t-1}$ depends solely on $x_t$, we can alter the known regions for $V$ as long as we keep the correct properties of the corresponding distribution, which is modeled by

$$p(x_t \mid x_0) = \mathcal{N}(\sqrt{1 - \bar{\beta}_t} x_0, \bar{\beta}_t I) \tag{47}$$

for $\bar{\beta}_t = 1 - \exp\left(-\int_0^t \beta_u du\right)$.

Thus, using the pretrained reverse SDE $dx_t = (-\frac{1}{2}\beta_t x_t - \beta_t s_{\theta^*}(t, v_t, r_t, \mid \mathcal{F}))dt + \sqrt{\beta_t}d\bar{W}_t$ to sample only the unknown region and equation 47 for the known regions, we get the following reverse SDE for $\{r_t\}_{0 \leq t \leq T}$.

$$dr_t = \left[-\frac{1}{2}\beta_t r_t - \beta_t(m^R \odot s_{\theta^*}(t, v_t, r_t, \mid \mathcal{F}))\right] dt + \sqrt{\beta_t}d\bar{W}_t, \tag{48}$$

where $v_t \sim \mathcal{N}(\sqrt{1 - \bar{\beta}_t}V, \bar{\beta}_t I)$ and $m^R \odot$ is the masking of output scores to collect only the values in unknown regions for $R$.

Comparing equation 48 with the true reverse SDE of $r_t$ for sampling from $p_t(r_t \mid V, \mathcal{F})$: $dr_t = \left[-\frac{1}{2}\beta_t r_t - \beta_t \nabla_{r_t} \log p(r_t \mid V, \mathcal{F}))\right] dt + \sqrt{\beta_t}d\bar{W}_t$, we can define the estimator for conditional score $\nabla_{r_t} \log p(r_t \mid V, \mathcal{F})$ appear in the latter as

$$s_{\theta^*}(t, r_t \mid V, \mathcal{F}) = m^R \odot s_{\theta^*}(t, v_t, r_t \mid \mathcal{F}), \quad \text{where} \quad v_t \sim \mathcal{N}(\sqrt{1 - \bar{\beta}_t}V, \bar{\beta}_t I) \tag{49}$$

where $m^R \odot$ is the masking of output scores to collect only the values in unknown regions for $R$, and it suffices the following:

$$s_{\theta^*}(t, r_t \mid V, \mathcal{F}) \approx \nabla_{r_t} \log p_t(r_t | V, \mathcal{F}). \tag{50}$$

### G.6 ALGORITHM FOR LINKERDPS

We describe alorithm of LinkerDPS in Algorithm 4, which is to adopt ancestral sampling (Ho et al., 2020a) with our reverse process discussed in Section 3.3. Similarly to prior work (Chung et al., 2024a), we choose step size $\{\xi_t\}_{t=1}^T$ to be $\xi_t = \frac{\xi'}{\|\nabla_{\hat{r}}\left(\phi_1(\hat{r}|\tilde{R})+\phi_2(\hat{r}|\tilde{E})\right)\|}$ with $\xi' = 0.01$.

---

**Algorithm 4** LinkerDPS

---

1: **INPUT** pretrained score network of PC-Aware model $s_{\theta^*}$, PC-Aware fragment condition $\mathcal{F}$, surrogate molecule $\tilde{G}$, $\phi_1(r \mid \tilde{R}) := \|\tilde{R} - r\|^2$, $\phi_2(r \mid \tilde{E}) := \sum_{1 \le i,j \le N} \mathbb{1}_{\tilde{E}_{i,j} \ne \mathbf{0}} \|r_i - r_j\|$, step size $\{\xi_t\}_{t=1}^T$, variance level $\{\bar{\beta}_t = 1 - \exp(-\int_0^t \beta_u du)\}_{t=1}^T$

2: $(\tilde{V}, \tilde{E}, \tilde{R}) \leftarrow \tilde{G}$

3: $V' \leftarrow \tilde{V}$

4: $v_0 \xleftarrow[\text{embedding}]{\text{continuous}} V'$

5: $r_T \sim \mathcal{N}(0, I)$                                                   ◁ run reverse process on $r$

6: **for** $t = T, \dots, 1$ **do**

7:     $\epsilon \sim \mathcal{N}(0, I)$

8:     $v_t \leftarrow \sqrt{1 - \bar{\beta}_t} v_0 + \sqrt{\bar{\beta}_t} \epsilon$

9:     $\hat{s} \leftarrow s_{\theta^*}(t, v_t, r_t \mid \mathcal{F})$              ◁ estimate conditional score $\nabla_{r_t} \log p_t(r_t \mid v_0)$

10:     $\hat{r} \leftarrow \frac{1}{\sqrt{1-\bar{\beta}_t}}(r_t + \bar{\beta}_t \hat{s})$               ◁ compute $\hat{r} \approx \mathbb{E}_{r_0 \mid r_t, V, \mathcal{F}}[r_0]$

11:     $z \sim \mathcal{N}(0, I)$

12:     $r'_{t-1} \leftarrow \frac{\sqrt{\alpha_t}(1-\bar{\alpha}_{t-1})}{1-\bar{\alpha}_t} r_t + \frac{\sqrt{\bar{\alpha}_{t-1}}\beta_t}{1-\bar{\alpha}_t} \hat{r} + \sqrt{\bar{\beta}_t} z$

13:     $r_{t-1} \leftarrow r'_{t-1} - \xi_t \nabla_{r_t}\left(\phi_1(r \mid \tilde{R}) + \phi_2(r \mid \tilde{E})\right)$    ◁ apply guidance from the $\tilde{E}, \tilde{R}$

14: **end for**

15: $R' \leftarrow r_0$

16: **return** $(V', R')$

---

# H ADDITIONAL EXPERIMENT ON GEOM-DRUG FRAGMENTS

We include experiments on synthetic fragments from GEOM-DRUG, showing consistent diversity–validity trade-offs. Results indicate that LinkerDPS generalizes across diverse fragment distributions.

Table 6: **GEOM-DRUG fragment results.** All metrics are percentages (%). F/A indicates the model category (F = PC-Free; A = PC-Aware; F+A = Hybrid). "V+HD/FG/BM" are task-specific property metrics combined with validity.

| Name | F/A | V+U | V+N | V+HD | V+FG | V+BM | Validity | Uniq (w/o V) | Novel (w/o V) |
|---|---|---|---|---|---|---|---|---|---|
| FFLOM | F | 29.39 | 28.30 | 6.95 | 21.65 | 15.87 | 30.02 | **95.07** | **91.73** |
| DeLinker | F | 40.52 | 26.18 | 8.29 | 19.51 | 20.32 | 52.22 | 75.90 | 52.31 |
| DiffLinker | A | 42.24 | 30.77 | 9.15 | 19.26 | 21.65 | 64.32 | 62.32 | 46.54 |
| HYBRIDLINKER(FFLOM) | F+A | 54.64 | **49.81** | **12.82** | **33.77** | **28.91** | 56.19 | 79.29 | 72.22 |
| HYBRIDLINKER(DeLinker) | F+A | **56.41** | 38.67 | 11.33 | 25.44 | 27.18 | **71.85** | 75.33 | 54.48 |

# I DESCRIPTOR OPTIMIZATION

In this experiment, we identify the molecule whose descriptor value is closest to a randomly sampled target value to assess the effectiveness of each generation algorithm. The target value is drawn from an approximated Gaussian distribution of the descriptors of reference molecules in the test dataset. We then select the molecule that minimizes the descriptor distance to the target value. This minimum distance serves as the evaluation metric, where a smaller value indicates that the algorithm better captures a diverse range of molecular properties. We consider five molecular descriptors: Ipc, MolLogP, MolWt, TPSA, and LabuteASA. Table 7 presents the average score of each algorithm across the fragments in the ZINC test dataset. The results highlight the strength of HybridLinkers, as they achieve the best or second-best scores across most descriptors.

Table 7: Comparison of algorithms based on descriptor optimization. The values represent the minimum MAE between the target value and values of generated samples for given fragment.

| Method | Ipc | MolLogP | MolWt | TPSA | LabuteASA |
|---|---|---|---|---|---|
| FFLOM | 0.216 | 0.135 | 0.133 | 0.145 | 0.154 |
| DeLinker | 0.218 | 0.085 | 0.138 | 0.114 | 0.141 |
| DiffLinker | 0.219 | 0.114 | 0.172 | 0.132 | 0.177 |
| 3DLinker | 0.217 | 0.101 | 0.124 | 0.107 | 0.135 |
| HybridLinker (FFLOM) | **0.209** | 0.087 | **0.117** | 0.099 | 0.139 |
| HybridLinker (DeLinker) | 0.213 | **0.064** | 0.127 | **0.076** | **0.134** |

# J  ABSOLUTE SCORES FOR DRUG-LIKENESS OPTIMIZATION

**Setup.**   For each fragment pair of ZINC test dataset, we compute Top-$k$ scores per model and then average across all 400 pairs (supporting Table 3). Higher is better for QED; lower is better for SA. The results show consistent gains from HybridLinker refinement.

Table 8: **Top-**1 **Drug-likeness.** Per-pair Top-1, averaged over 400 pairs.

| Model | QED (↑) | SA (↓) |
|---|---|---|
| DiffLinker | 0.757 | 2.703 |
| 3DLinker | 0.766 | 2.667 |
| FFLOM | 0.748 | 2.781 |
| HYBRIDLINKER(FFLOM) | 0.769 | 2.686 |
| DeLinker | 0.764 | 2.641 |
| HYBRIDLINKER(DeLinker) | **0.770** | **2.626** |

Table 9: **Top-**5 **Drug-likeness.** Per-pair Top-5, averaged over 400 pairs.

| Model | QED (↑) | SA (↓) |
|---|---|---|
| DiffLinker | 0.729 | 2.772 |
| 3DLinker | 0.733 | 2.731 |
| FFLOM | 0.696 | 2.918 |
| HYBRIDLINKER(FFLOM) | 0.741 | 2.751 |
| DeLinker | 0.738 | **2.648** |
| HYBRIDLINKER(DeLinker) | **0.748** | 2.627 |

Table 10: **Top-**10 **Drug-likeness.** Per-pair Top-10, averaged over 400 pairs.

| Model | QED (↑) | SA (↓) |
|---|---|---|
| DiffLinker | 0.708 | 2.784 |
| 3DLinker | 0.708 | 2.739 |
| FFLOM | 0.654 | 2.969 |
| HYBRIDLINKER(FFLOM) | 0.717 | 2.780 |
| DeLinker | 0.712 | **2.621** |
| HYBRIDLINKER(DeLinker) | **0.726** | 2.592 |

**Takeaway.**   Across $k \in \{1, 5, 10\}$, HybridLinker consistently improves QED and reduces PLogP relative to its PC-Free counterparts, while maintaining competitive SA—supporting the claim that refinement yields higher drug-likeness without sacrificing synthesizability.

## K  SENSITIVITY OF HYBRID PERFORMANCE TO THE CHOICE AND QUALITY OF THE SURROGATE PC-FREE MODEL

HybridLinker is robust to the choice of surrogate PC-Free model, yet benefits from stronger surrogates. On our primary metric family (*Diversity with Validity*), HybridLinker(FFLOM) and HybridLinker(DeLinker) are comparable; however, the pure-validity metric (ignoring diversity) is markedly higher for HybridLinker(DeLinker), reflecting DeLinker's stronger validity relative to FFLOM. We further quantify the relationship between surrogate quality and output quality on the ZINC test set using both DeLinker and FFLOM as surrogates. Across all settings, HybridLinker consistently surpasses single-model baselines (FFLOM, DeLinker, 3DLinker, DiffLinker) on *Diversity with Validity*.

**Energy correlation (diagnostic).**   To probe structural plausibility, we report the energy of the surrogate output (*Original*) and the refined output (*New*). Although energy is not a target metric, refined outputs correlate with surrogate quality.

Table 11: **Energy analysis with DeLinker as surrogate.** Energies in kcal/mol. $\Delta = \text{New} - \text{Original}$. Percentile bins are computed over the *Original* energy distribution.

| Percentile Bin | Mean Original | Mean New | Mean $\Delta$ |
|---|---|---|---|
| 0–10 | 108.23 | 153.93 | 45.70 |
| 10–20 | 121.68 | 163.56 | 41.88 |
| 20–30 | 134.19 | 169.82 | 35.63 |
| 30–40 | 147.45 | 179.50 | 32.04 |
| 40–50 | 161.70 | 183.96 | 22.26 |
| 50–60 | 179.09 | 197.31 | 18.22 |
| 60–70 | 199.65 | 206.84 | 7.19 |
| 70–80 | 224.66 | 220.46 | -4.20 |
| 80–90 | 261.49 | 244.28 | -17.22 |
| 90–100 | 335.09 | 289.79 | -45.30 |
| **Overall (mean $\pm$ sd)** | $187.32 \pm 66.77$ | $200.94 \pm 39.40$ | $13.62 \pm 27.44$ |

*Pearson r* (Original vs. New): 0.6220.

Table 12: **Energy analysis with FFLOM as surrogate.** Energies in kcal/mol. $\Delta = \text{New} - \text{Original}$. Percentile bins are computed over the *Original* energy distribution.

| Percentile Bin | Mean Original | Mean New | Mean $\Delta$ |
|---|---|---|---|
| 0–10 | 114.74 | 175.88 | 61.14 |
| 10–20 | 133.17 | 179.96 | 46.79 |
| 20–30 | 150.82 | 184.81 | 33.99 |
| 30–40 | 168.25 | 190.00 | 21.75 |
| 40–50 | 189.07 | 200.51 | 11.44 |
| 50–60 | 213.82 | 212.98 | -0.85 |
| 60–70 | 242.39 | 221.00 | -21.40 |
| 70–80 | 283.91 | 239.99 | -43.92 |
| 80–90 | 337.31 | 258.49 | -78.82 |
| 90–100 | 414.62 | 288.22 | -126.41 |
| **Overall (mean $\pm$ sd)** | $224.81 \pm 91.14$ | $215.18 \pm 35.24$ | $-9.63 \pm 55.98$ |

## L  COMPONENT-WISE ABLATION OF LINKERDPS

The refinement process using LinkerDPS plays a central role in HybridLinker. To assess the contribution of each component in $p(\tilde{G} \mid V, R)$—namely, the likelihood of atoms, bonds, and conformation, each providing three types of guidance—we conduct an ablation study in Table 13. Variants of LinkerDPS (A-DPS) are tested by selectively removing specific likelihood components. Additionally, we introduce a version where the likelihood of atoms is modeled as a continuous distribution using a Gaussian kernel:

$$p(\tilde{v} \mid v) \propto \kappa(\tilde{v}, v), \tag{51}$$

which allows for a continuous representation of atomic properties. To implement this, we employ the following reverse process:

$$dx_t = \left[ f(x_t, t) - g^2(t)\nabla_{x_t} \log p_t(x_t \mid \tilde{V}, \tilde{E}, \tilde{R}) \right] dt + g(t)d\bar{W}_t, \tag{52}$$

where $x_0 = (V, R)$. The LinkerDPS approximation for $p_t(\tilde{V}, \tilde{E}, \tilde{R} \mid x_t)$ is straightforward to adapt, and we apply this approximation to compute the score in equation 52.

Table 13 reveals that even variants using a single type of guidance outperform baseline methods. Performance improves further when combining two types, with the best results achieved when all three guidance components are incorporated. The marginal gains from additional guidance suggest that each component provides complementary yet overlapping information about molecular topology. Notably, relaxing the condition on atoms reduces molecular diversity, while the strict constraint on atoms ensures the preservation of surrogate atoms, reinforcing its role in maintaining molecular consistency. While validity slightly improves in the absence of inpainting, this is likely due to the repeated generation of certain valid molecules, as reflected in the decline of diversity metrics that count only valid molecules (V+U and V+N).

Table 13: Ablation study on likelihood type in LinkerDPS. The rightmost two columns represent **diversity of valid molecules**. The likelihood components used in each variant are denoted with A (Atom), E (Bond), and R (conformation).

| Guidance Type | Unique | Novel | Valid | V+U | V+N |
|---|---|---|---|---|---|
| E | 57.17 | 33.55 | **76.89** | 49.82 | 29.30 |
| R | 57.68 | 33.47 | **76.89** | 49.93 | 28.97 |
| A | 65.10 | 41.79 | 68.39 | 53.59 | 33.71 |
| A + E | 67.20 | 44.10 | 69.23 | 54.59 | 35.08 |
| A (DPS) + E + R | 61.61 | 37.76 | 75.65 | 52.12 | 31.61 |
| A + E + R (Ours) | **68.45** | **44.69** | 69.14 | **55.10** | **35.17** |

## M    EXPERIMENTS ON HYPERPARAMETER SETUP

HYBRIDLINKER is training-free and has only three scalar hyperparameters: the bond-term weight $\omega_b$, the coordinate-term weight $\omega_c$, and the DPS strength $\lambda_{\text{DPS}}$. Our measurement objective combines likelihood components and DPS guidance as

$$\mathcal{L}_{\text{meas}} = \log p_{\text{atom}} + \omega_b \log p_{\text{bond}} + \omega_c \log p_{\text{coord}} + \lambda_{\text{DPS}} \, \Phi_{\text{DPS}} \,, \tag{53}$$

where $p_{\text{atom}}$, $p_{\text{bond}}$, and $p_{\text{coord}}$ denote the atom, bond, and conformation/coordinate likelihoods, respectively, and $\Phi_{\text{DPS}}$ is the DPS guidance term. As shown in Algorithm 4 (Sec. F.6), $\lambda_{\text{DPS}}$ scales guidance strength, while $\omega_b, \omega_c$ set the relative contribution of bond vs. coordinate terms. For simplicity, we use a unit atom weight (i.e., an implicit $\omega_a{=}1$), which we omitted in the pseudocode.

We swept $\{\lambda_{\text{DPS}}, \omega_b, \omega_c\}$ over the values reported below and found performance to be robust. Ablations use HybridLinker(FFLOM) on ZINC with default pretrained configurations.

Table 14: **Hyperparameter sensitivity on ZINC with HybridLinker(FFLOM).** Each triple denotes $(\lambda_{\text{DPS}}, \omega_b, \omega_c)$. All metrics are percentages (%).

| $(\lambda_{\text{DPS}}, \omega_b, \omega_c)$ | V+U | V+N | Validity | Uniqueness | Novelty |
|---|---|---|---|---|---|
| (0.10, 0.010, 0.005) | 54.08 | 34.32 | 68.13 | 66.31 | 42.82 |
| (0.10, 0.010, 0.015) | 53.85 | 34.10 | 67.45 | 67.16 | 43.79 |
| (0.10, 0.005, 0.010) | 53.38 | 33.84 | 67.68 | 66.25 | 42.93 |
| (0.10, 0.015, 0.010) | 54.14 | 34.08 | 67.97 | 67.60 | 43.75 |
| (0.01, 0.010, 0.010) | 53.71 | 34.07 | 67.44 | 67.18 | 43.38 |
| (0.05, 0.010, 0.010) | 53.79 | 33.96 | 67.82 | 67.08 | 43.26 |
| (0.15, 0.010, 0.010) | 54.17 | 34.39 | 67.76 | 67.62 | 43.88 |

# N INFERENCE TIME AND COMPUTATIONAL COST

Because HybridLinker runs a two–stage pipeline (PC–Aware + PC–Free) at inference, it incurs higher wall-clock time than single-stage methods. In our measurements, per–molecule runtime is 1.70–1.78 s for HybridLinker compared to 0.13–1.08 s for single-stage baselines—i.e., typically $\sim 1.5$–$3\times$ slower than diffusion baselines, with larger gaps against the fastest autoregressive models. We argue this is a reasonable trade-off for drug design workflows (which are not real-time) given the substantial gains in both diversity and validity reported in the main results.

Table 15: **Per-molecule inference time.** "Pre." indicates a pretrained model used in inference only. Times are in seconds (s).

| Method | Component(s) | Time (s) |
|---|---|---|
| FFLOM | FFLOM | 0.132 |
| DeLinker | DeLinker | 0.240 |
| 3DLinker | 3DLinker | 1.080 |
| DiffLinker | DiffLinker | 0.565 |
| HYBRIDLINKER (FFLOM) | DiffLinker (Pre.) + FFLOM (Pre.) | 1.700 |
| HYBRIDLINKER (DeLinker) | DiffLinker (Pre.) + DeLinker (Pre.) | 1.780 |

**Practicality.** The two-stage cost is amortizable in practice: both stages are embarrassingly parallel across candidates/seeds, and HybridLinker is training-free with only a few scalar controls, minimizing tuning overhead.

## O  ADDITIONAL DISCUSSION ON EXPERIMENTAL RESULTS

**Impact of Balancing Diversity and Validity in Drug Discovery**   Balancing diversity and validity is a fundamental challenge in drug discovery, as it directly determines the success of downstream applications. Our experimental results further reinforce this: HybridLinker's strength in Table 2 translates to superior performance in Table 3. The findings reveal that models excelling in only one aspect—either generating diverse but invalid molecules or producing valid but overly constrained structures—lack practical utility in drug design. This highlights the necessity of striking a balance between diversity and validity for practical applications, establishing HybridLinker as a crucial framework for advancing molecular generation in drug discovery.

**Impact of surrogate quality on HybridLinker's performance**   Comparing the two HybridLinker implementations in Table 2, each utilizing FFLOM and DeLinker as surrogate generators, highlights the impact of surrogate quality on HybridLinker's performance. FFLOM exhibits higher Uniqueness and Novelty than DeLinker, and accordingly, HybridLinker(FFLOM) achieves superior scores in these metrics compared to HybridLinker(DeLinker). Conversely, HybridLinker(DeLinker) demonstrates higher validity, reflecting DeLinker's tendency to generate more valid molecules than FFLOM. A similar trend is observed in drug-likeness optimization results presented in Table 3 and Table 7, where DeLinker-produced molecules generally exhibit better drug-likeness, which corresponds to HybridLinker(DeLinker) achieving stronger drug-likeness scores. Likewise, in the molecular descriptor optimization task, implementations using surrogates that perform well for specific descriptors tend to show better optimization results for those properties. Furthermore, the effectiveness of LinkerDPS in molecule refinement is evident in Table 2. While HybridLinker(DeLinker) retains higher validity than HybridLinker(FFLOM), the validity gap between them significantly narrows. This suggests that LinkerDPS successfully refines invalid molecules into valid ones, effectively mitigating the initial disparity between FFLOM and DeLinker.

## P    RELATED WORKS: MOLECULE GENERATION AND GUIDANCE ON DIFFUSION MODELS

Here, we summarize the related works in molecule generation domain.

Table 16: Comparison of molecular generation tasks based on inputs and outputs. It reveals that our setup, linker generation, is a general formulation of fragment-based molecular generation. $G$ represents a complete molecule represented as a 3D graph, while $F$, another 3D graph, is a partial molecule with $|F| \ll |G|$ in most cases. $M$ denotes the binding pocket of target proteins, providing structural constraints. $\mathcal{T}$ refers to the bonding topology of a molecule, defined by atom types and chemical bonds. $\mathbf{T}$ represents an arbitrary transformation in SE(3) (e.g., rotations and translations).

| Task | Linker Gen. (Our Setup) | 2D Linker Gen. | Scaffold Hop. | PROTAC Design |
|---|---|---|---|---|
| Input | $G_1, G_2$ | $\mathcal{T}_1, \mathcal{T}_2$ | $F_1, F_2$ | $G_1, G_2$ |
| Output | $G' \mid G_1, G_2 \subset G'$ | $\mathcal{T}' \mid \mathcal{T}_1, \mathcal{T}_2 \subset \mathcal{T}'$ | $G' \mid F_1, F_2 \subset G'$ | $G' \mid \mathbf{T}_1[G_1], \mathbf{T}_2[G_2] \subset G'$ |
| Task | Target-Aware Drug Design | Fragment Growing | De Novo Gen. | Conformation Gen. |
| Input | $G_1, G_2, M'$ | $G^\star$ | None | $\mathcal{T}^\star$ |
| Output | $G' \mid G_1, G_2 \subset G, M'$ | $G' \mid G^\star \subset G'$ | $G'$ | $G' = (\mathcal{T}^\star, R') \mid \mathcal{T}^\star$ |

**Fragment-Based Drug Design.**    Fragment-Based Drug Design (FBDD)(Jin et al., 2023; Igashov et al., 2024; Torge et al., 2023; Imrie et al., 2020; Huang et al., 2022) is a drug discovery approach that utilizes small molecular fragments and optimizes them into larger, more potent drug candidates. Deep learning-based FBDD encompasses a variety of tasks, each distinguished by its learning objective. **Linker Generation** (Igashov et al., 2024; Huang et al., 2022) is a fundamental task in FBDD, where two molecular fragments are connected to form a complete molecule. Similarly, **Topology Linker Generation** (Jin et al., 2023; Imrie et al., 2020; Zhang et al., 2024) focuses on linking the topological graphs of fragments to generate a complete molecular topology graph. **Scaffold Hopping**(Torge et al., 2023; Zhou et al., 2024) involves replacing the core structure of a given molecule while preserving its biological activity. **PROTAC Design**(Guan et al., 2024; Kao et al., 2023) focuses on generating molecules that incorporate fragment linkers with flexible rotation and translation in 3D space. **Fragment Growing**(Maziarz et al., 2024; Ghorbani et al., 2023) expands single small molecular fragment into larger drug-like structure. Our work specifically addresses Linker Generation, tackling the critical trade-off between diversity and validity observed in existing models. Table 16 Summarizes the tasks in FBDD along with two standard molecular generation tasks—De Novo Generation and Conformation Generation.

**Molecule Generation.**    Molecule generation(Hoogeboom et al., 2022; Vignac et al., 2023; Wang et al., 2024b; Liu, 2021; Madhawa et al., 2019; Schneuing et al., 2024) based on deep learning plays a crucial role in drug discovery and is broadly categorized into **De Novo Molecule Generation**(Peng et al., 2023; Geng et al., 2023; Vignac et al., 2023; Jo et al., 2024), **Fragment-Based Drug Design**(Jin et al., 2023; Igashov et al., 2024; Torge et al., 2023; Imrie et al., 2020; Huang et al., 2022), **Target-Aware Drug Design**(Guan et al., 2023; Corso et al., 2023a; Schneuing et al., 2024; Peng et al., 2022), and **Conformer Generation**(Xu et al., 2022; Kim et al., 2024; Jing et al., 2023; Xu et al., 2021), based on their input and output formulations. Molecule generation can also be classified into three sub-tasks: **Topology Generation**(Shi et al., 2020; Geng et al., 2023; Jin et al., 2023; Jo et al., 2024; Ji et al., 2020; Wang et al., 2024b;a), **Point Cloud Generation**(Hoogeboom et al., 2022; Xu et al., 2023; Igashov et al., 2024; Guan et al., 2023; Schneuing et al., 2024), and **3D Graph Generation**, where deep learning models learn the distributions of molecular bonding topology, spatial coordinates, and 3D molecular graph representations, respectively. Our work falls under Fragment-Based Drug Design, introducing a hybrid approach that leverages pretrained models for topology and point cloud generation in a training-free manner.

**Guidance on Diffusion Models.**    Diffusion models(Ho et al., 2020b; Song et al., 2022; Rombach et al., 2022; Ho et al., 2022; Jo et al., 2024; Hoogeboom et al., 2022; Vignac et al., 2023) have demonstrated exceptional performance across various generative tasks, including image, video, graph, and molecular generation. A recent advancement in diffusion models is **conditional generation**(Ho & Salimans, 2022; Chung et al., 2024a;b; Dhariwal & Nichol, 2021), which enables sampling from a

conditional distribution based on desired properties. This is achieved by incorporating a guidance term into the backward diffusion process. To compute the guidance term, **Classifier Guidance** (Dhariwal & Nichol, 2021) employs a classifier trained to estimate the likelihood of a given property, while **Classifer-Free Guidance** (CFG) (Dhariwal & Nichol, 2021) replaces the classifier with a conditional diffusion model. **CFG++** (Chung et al., 2024b) is designed to mitigate off-manifold sampling issues, and **Diffusion Posterior Sampling** (DPS) (Chung et al., 2024a) was developed to solve nonlinear noisy inverse problems. In this paper, we introduce the first DPS-based method for guiding diffusion models in molecular point cloud generation using molecular topology. Our approach introduces a novel energy-based function that effectively bridges topological and spatial molecular representations.

