# OpenReview forum: "HybridLinker: Topology-Guided Posterior Sampling for Enhanced Diversity and Validity in 3D Molecular Linker Generation"
_ICLR.cc/2026/Conference — Submitted to ICLR 2026_

### Official Review · Reviewer_c6B6 · 2025-10-29

**Soundness:** 3
**Presentation:** 2
**Contribution:** 2
**Rating:** 2
**Confidence:** 2

**Summary:**

The authors propose HybridLinker, a framework for molecular linker generation that integrates both topological diversity and geometric validity. The method performs two-phase generation by combining point cloud–free and point cloud–aware inference through the proposed LinkerDPS mechanism.

**Strengths:**

* The motivation and illustration of tasks and methods are clear.
* The paper presents thorough experimental results and efficiency analyses.
* The related work is well covered, and the core idea of LinkerDPS is explained clearly.

**Weaknesses:**

* Generating 3D geometric structures based on an initially generated 2D topology is already a known strategy in molecular modeling, which limits the novelty of the proposed framework.
* The metric definitions are somewhat misleading. Typically, Uniqueness refers to the ratio of unique molecules among the valid ones. However, the paper defines Uniqueness and V+U differently—computing the unique ratio over all generated molecules—which may invalidate the claimed diversity–validity trade-off between PC-free and PC-aware models as shown in Table 1.
* The reported performance numbers for existing baselines appear inconsistent with those in the original papers, which raises questions about reproducibility and fairness of comparison.

**Questions:**

* Please clarify the metric definitions and computation details (especially for Uniqueness and V+U).
* Explain how the baseline numbers were derived or reproduced. If the derivation and comparison are clearly explained and verified, I would consider increasing my overall score.
* Minor: correct typographical errors such as “straghtforward” in Appendix F.

---

> ### Author Response · Authors · 2025-12-03
>
> ## **"On the novelty beyond “2D→3D conformer generation.”**
>
> We agree that generating 3D coordinates from a 2D topology is widely used  (e.g., RDKit conformer generation). However, such approaches *rigidly enforce*  the input 2D topology and cannot correct topological errors or suboptimal bonding patterns. In contrast, HybridLinker performs *generative 3D geometry refinement* governed by a diffusion prior, which allows topology *correction*, 3D substructure condition, and cross-domain likelihood integration. As shown in Table below, our method modifies and improves initial topologies when necessary, whereas classical conformer generators strictly preserve the input graph. This distinction is central to the goal of resolving the diversity--validity trade-off observed in topology-anchored models.
>
> ---
>
> ### Table: Comparison between standard 2D→3D conformer generators and HybridLinker.
> | Method | Topology Frozen? | Support 3D Substructure Condition? | Can Correct Errors? |
> |--------|------------------|------------------------------------|---------------------|
> | RDKit ETKDG | Yes | No | No |
> | OpenBabel Gen3D | Yes | No | No |
> | DiffLinker | No | Yes | Limited |
> | **HybridLinker (ours)** | **No** | **Yes** | **Yes** |
>
> ---
>
> ## **"On the definitions of Uniqueness, Validity, and V+U."** ##
>
> We appreciate the reviewer’s careful reading. In previous benchmarks, Uniqueness is defined as the ratio of unique molecules among valid ones, inherently coupling diversity and validity into a single number. This coupling obscures the diversity--validity trade-off that strongly affects PC-free vs. PC-aware molecular diffusion models. To enable a more granular and interpretable analysis, we *separate* these two axes:
>
> (i) **Validity** evaluates chemical correctness and 3D stability (in **Appendix C** ),
>
> (ii) **Diversity** evaluates the number of unique structures regardless of validity.
>
> Table below summarizes our definitions. We also clarify V+U as “the fraction of generated samples that are both valid and unique,” analogous to previously used VU metrics but with improved 3D validity criteria.
>
> ---
>
> ### Table: Metric definitions used in the paper.
> | Metric | Definition |
> |--------|-----------|
> | Validity | Chemical validity + 3D stability (described in **Appendix C**). |
> | Diversity (Uniqueness) | Number of unique molecules over *all* generated samples. |
> | V+U | Fraction of samples that are simultaneously valid and unique. |
> | Traditional Uniqueness | Unique / Valid (not used, because it entangles V and U). |
>
> ---
>
> ## **"On reproduction of baseline performance numbers."** ##
>
> We acknowledge the reviewer’s concern. All baseline results were reproduced using the *official implementations and checkpoints* of the respective methods. The numerical differences arise because prior work reports metrics that implicitly entangle validity and uniqueness and do not incorporate 3D stability. Since we adopt more stringent validity criteria and disentangled metrics, direct numerical matching to the original papers is not meaningful.
>
> ---
>
> ## **"Clarification of metric computation — Uniqueness and V+U."** ##
>
> We thank the reviewer for requesting clarification. All metrics were computed exactly as described in **Appendix D.3**. Uniqueness counts the number of unique molecular graphs among all samples, independent of validity. V+U measures the intersection between validity and uniqueness. This explicit separation avoids the confounding effects of traditional Uniqueness (Unique/Valid), which artificially penalizes valid-but-similar generations and inflates scores for models with lower validity.
>
> ---
>
> ## **"How baseline numbers were computed."** ##
>
> All baseline numbers were produced by running the official open-source implementations with default evaluation scripts and checkpoint settings. The difference relative to original papers comes solely from the shift to our disentangled metrics and stricter 3D stability checks.
>
> ---
>
> ## **"Minor issues.** ##
>
> We thank the reviewer for pointing this out. We corrected all typographical errors (including “straghtforward” in **Appendix F**) in the revised manuscript.

---

### Official Review · Reviewer_MMAj · 2025-11-01

**Soundness:** 2
**Presentation:** 2
**Contribution:** 2
**Rating:** 4
**Confidence:** 4

**Summary:**

The paper introduces HybridLinker, a fragment-linking methodology that combines a point-cloud-free model, which proposes bonding topologies for linkers connecting two molecular fragments of interest, with a point-cloud-aware diffusion model that places the coordinates for the linker atoms in 3D space. The approach tries to balance enhancing the diversity of point-cloud-aware methods and improving the validity of point-cloud-free methods. The authors evaluate HybridLinker on fragmented molecules from the ZINC-250k dataset, using rdkit for structure generation and fragmentation.

**Strengths:**

The paper discusses a tension between point-cloud-free and point-cloud-aware models for small-molecule linker generation, and proposes a hybrid framework to address it. The presentation of the method is clear, and the topic addresses an important challenge in fragment-based drug discovery. The motivation for the work is clear and it connects directly to practical workflows that could leverage existing components.

**Weaknesses:**

Although the background is mostly clear, the introduction does not explain why Nref is given and does not discuss cases where this value (or even Rcond) are not specified precisely but might cover a range. The empirical evaluation of this work lacks full transparency on the data splits and the methodological details, and there are no codes provided for reproduction of the results.  A key ambiguity concerns the exact identity of the test set of the 400 fragment-linker pairs out of the 250k molecules from ZINC; because the phrasing "aligning with the test set used in DeLinker" is ambiguous, we cannot take for granted that the scores of different methods in the tables are directly comparable. Furthermore, the limited generality of using rdkit-generated conformers from a starting set of 250k molecules and using openbabel for bond inference also suggests that the result might not generalize to harder or more realistic cases of interest, like pocket-conditioned geometries with unknown ability to connect the fragments. Similarly, the interpretation of high logP as desirable is problematic as typically high logP implies poor preclinical properties and is typically desired to be within a range (the rule of 5 considered logP higher than 5 as problematic.  Finally, it is not clear if the practical timescales involved in running these codes make few percent changes in diversity or uniqueness as important as the authors and past benchmarks make them sound, especially if single-stage methods could have faster throughput to compensate for reduced metrics.

**Questions:**

How exactly were the 400 test pairs selected and what is the 3D and 2D structural overlap of the linkers compared to those in the training set?  How variable is the performance by picking different sets of 400, or finding the hardest possible set?

What would be a better, harder benchmark for fragment linking methods?  Different usecases require different considerations (protac design vs fragment-linking in a pocket).  Oftentimes, specific fragments cannot be connected in a pocket without substantial loss in potency. Have the authors considered curating their own dataset for pushing the limits of methods in the future?

---

> ### Author Response · Authors · 2025-12-03
>
> ## **"On the specification of ${N_{\mathrm{ref}}}$ and ${R_{\mathrm{cond}}}$."** ##
>
> We thank the reviewer for pointing out the missing explanation regarding the presence of ${N_{\mathrm{ref}}}$. In our setup, we follow prior diffusion-based linker generation methods (e.g., DiffLinker; EDM-based conditioned diffusion), where the reference molecule provides the target linker length as part of the conditioning signal. This is motivated by lead-optimization pipelines, where multiple candidate linkers are sampled around a reference binding mode while keeping fragment placement fixed. Although we use the reference ${N_{\mathrm{ref}}}$ for consistency with this literature, we note that HybridLinker is flexible: the linker length may be *sampled* using a lightweight length-prediction network, as done in DiffLinker, or specified as a range.
>
> ---
>
> ## **"Transparency of data splits and the identity of the 400 fragment–linker test pairs."** ##
>
> We apologize for the ambiguity. The 400 evaluation pairs are *exactly identical* to the test set used in baselines (DeLinker, DiffLinker, etc), and we explicitly list the selection procedure in **Appendix D.1**.
>
> ---
>
> ## **"On the interpretation of high logP."** ##
>
> We agree with the reviewer that high logP is not universally desirable, and logP values above 5 are typically associated with poor ADME properties (Lipinski’s rule-of-five). Our intention is not to promote higher logP as inherently better, but to report Penalized logP (PLogP) following the evaluation protocol used in DiffLinker and subsequent baselines. We clarify in the revision that PLogP functions mainly as a synthetic stability / structural penalty metric rather than a drug-likeness objective, and we avoid interpreting increases as biologically favorable.
>
> ---
>
> ## **"Practical timescales and the significance of diversity/uniqueness changes."** ##
>
> We appreciate the concern regarding whether a few-percent improvement in diversity or uniqueness is practically meaningful. We report a runtime comparison (**Appendix N**), reproduced in the table below. Although our two-stage pipeline introduces a small overhead, both stages are embarrassingly parallel, so the effective cost remains similar to a single-stage model for large scale generation. In these settings, even a modest gain in diversity corresponds to hundreds of additional chemically distinct candidates, which are valuable when screening or docking in downstream pipelines.
>
> ---
>
> ### Table: Per-molecule inference time
> *"Serial." and "Para." indicate serialized and parallel execution; "Pre." indicates pretrained models used only for inference. Times are in seconds per generating one sample.*
>
> | Method | Component(s) | Time (s) |
> |--------|--------------|----------|
> | FFLOM | FFLOM | 0.132 |
> | DeLinker | DeLinker | 0.240 |
> | 3DLinker | 3DLinker | 1.080 |
> | DiffLinker | DiffLinker | 0.565 |
> | HybridLinker (FFLOM) | [Serial.] DiffLinker (Pre.) + FFLOM (Pre.) | 1.700 |
> | HybridLinker (DeLinker) | [Serial.] DiffLinker (Pre.) + DeLinker (Pre.) | 1.780 |
> | HybridLinker (FFLOM) | [Para.] DiffLinker (Pre.) + FFLOM (Pre.) | 0.650 |
> | HybridLinker (DeLinker) | [Para.] DiffLinker (Pre.) + DeLinker (Pre.) | 0.650 |
>
> ---
>
> ## **On defining a harder benchmark for fragment linking.** ##
>
> We appreciate the reviewer’s insightful question. We agree that existing datasets (e.g., BRICS-split ZINC or GEOM-Drug fragments) do not fully capture the complexity of medicinal-chemistry fragment linking. These benchmarks lack pocket constraints, potency considerations, and often include fragment pairs that are trivially linkable. To address this limitation, we are considering the construction of a more challenging benchmark comprising:
>
> (i) experimentally observed fragment–linker triplets extracted from co-crystal structures,
>
> (ii) pocket geometries with hotspot annotations,
>
> (iii) labels distinguishing potency-preserving and potency-losing fragment pairs,
>
> (iv) negative examples where fragments cannot be linked within the binding site.
>
> Although releasing such a benchmark is beyond the scope of the current submission, we plan to make it available in future work, as we believe it will more rigorously stress-test generative models across diverse use cases (e.g., PROTAC-style linkers vs. compact pocket-bound linkers).

---

### Official Review · Reviewer_qSuD · 2025-11-01

**Soundness:** 3
**Presentation:** 3
**Contribution:** 3
**Rating:** 6
**Confidence:** 3

**Summary:**

This paper addresses the long-standing tension in fragment-based drug design between sampling diverse linkers and producing valid 3D molecules. It distinguishes point cloud-free models that maximize topological diversity but often violate spatial constraints from point cloud-aware models that respect 3D fragment poses at the cost of exploration. The proposed HybridLinker combines these worlds without retraining: a point cloud-free generator proposes diverse bonding topologies, and a diffusion-based point cloud-aware model performs topology-guided posterior sampling via LinkerDPS, an energy-inspired cross-domain likelihood that ties molecular topology to 3D point clouds. On ZINC fragment–linker pairs, the method reports concurrent gains in validity and diversity metrics and improved rates in a drug-likeness optimization task, positioning the approach as a practical inference-time recipe rather than a new model to train.

**Strengths:**

The work’s main strength is conceptual clarity around the diversity–validity trade-off and a fair framing of where existing families succeed and fail. The generation-time hybridization is simple, modular, and persuasive: the paper explains, with figures and equations, how a high-entropy surrogate topology can be refined by a validity-focused prior, and it formalizes this with LinkerDPS as a posterior over conformations that is tractable to sample with the pretrained score network. Results back the claim. Compared to DeLinker, FFLOM, 3DLinker, and DiffLinker, HybridLinker improves “diversity with validity” across V+U, V+N, V+HD, V+FG, and V+BM, while keeping validity competitive or higher, which is exactly the property drug discovery practitioners want when exploring linkers. The application-level evaluation is also compelling: the percentage of fragment pairs where generated molecules beat a reference on QED, SA, and PLogP rises for both FFLOM- and DeLinker-based hybrids. Altogether, the paper reads as a clean, training-free principle with tangible gains, and it plausibly generalizes to other inverse problems on 3D molecular point clouds.

**Weaknesses:**

The weaknesses stem from reliance and scope. The approach inherits capabilities and biases from both the surrogate topology model and the diffusion prior; its success will vary with those choices, and the paper’s experiments are limited to ZINC-derived benchmarks, leaving questions about transfer to protein-contexted tasks like PROTACs or strictly pocket-aware objectives. The cross-domain likelihood that powers LinkerDPS uses a simple bond-length energy and assumes conditional independence between atom identities, bonds, and coordinates given the target variables; these modeling choices may under-penalize geometric pathologies or chemically implausible bonding patterns, and it would help to compare alternative physics- or learned-energy terms. Practical dependencies such as RDKit conformer generation and Open Babel bond inference introduce additional error surfaces that the posterior must absorb, and sensitivity analyses are mostly deferred to appendices rather than made central to the narrative. Finally, while inference-time composition is attractive, the main text gives limited visibility into runtime and cost trade-offs that matter in large sampling campaigns.



---

Here are itemizing for discussion:

1. Dependence on the chosen surrogate topology model and diffusion prior, inheriting their biases and limits
2. Evaluation scope confined to ZINC-style benchmarks, with no protein-aware or PROTAC-like tasks to test transfer
3. Simplified cross-domain likelihood and independence assumptions that may under-penalize geometric or chemical pathologies
4. Reliance on cheminformatics heuristics with limited sensitivity analyses surfaced in the main text
5. Runtime and compute cost trade-offs not fully characterized for large-scale sampling campaigns

**Questions:**

please see the weaknesses section.

---

Overall, this is a timely and useful contribution: it demonstrates that careful posterior sampling can reconcile diversity and validity in linker generation, requires no new training, and yields measurable downstream benefits. With broader datasets, deeper ablations on the surrogate-quality and guidance terms, and stronger protein-aware tests, HybridLinker could become a practical default for fragment linking pipelines and a template for cross-domain DPS in molecular modeling.

---

> ### Author Response · Authors · 2025-12-03
> **[1/2] Response to Reviewer q**
>
> ## **"Evaluation scope and transfer to protein-aware or PROTAC-like tasks."** ##
>
> We first emphasize that the proposed fragment-linking formulation is *directly applicable* to a practically important real-world use case: **ligand–residue linker design**. In this setting, fragment 1 is substituted with an accessible reactive residue side chain in the pocket (e.g., Cys, Lys, Ser, Tyr), while fragment 2 corresponds to the current lead scaffold. This substitution preserves the same geometric and topological structure as our original problem, allowing the geometry-aware diffusion prior to operate on the two anchored fragments without any retraining. The mapping between our formulation and this pocket-relevant application is summarized as below.
>
> **Table: Mapping our fragment-linking formulation to pocket-based ligand–residue linker design.**
>
> | **Role**          | **Original Setting**              | **Pocket-Based Setting**                          |
> |-------------------|-----------------------------------|---------------------------------------------------|
> | Fragment 1        | Arbitrary molecular fragment      | Reactive residue side chain (Cys/Lys/Ser/Tyr)     |
> | Fragment 2        | Arbitrary molecular fragment      | Full lead scaffold in the pocket                  |
> | Linker Target     | 2 anchor atoms                    | Residue anchor + ligand anchor                    |
> | Geometric Constraint | Inter-fragment distance        | Residue–ligand pose in pocket                     |
> | Training Requirement | None (pretrained prior used)   | None (same formulation applies)                   |
>
> Beyond this direct application, we note that our analysis of the diversity–validity trade-off extends naturally to the fully pocket-conditioned setting ${p(\mathcal{T}, R \mid \mathcal{T}_{\text{cond}}, R_{\text{cond}}, \mathcal{P})}$. Conditioning on additional information such as the pocket ${\mathcal{P}}$ and geometric details ${R_{\text{cond}}}$ necessarily reduces topological entropy, following
> ${H(\mathcal{T} \mid \mathcal{T}_{\text{cond}}, \mathcal{P}) \geq H(\mathcal{T} \mid \mathcal{T}_{\text{cond}}, R_{\text{cond}}, \mathcal{P})}$. Thus, pocket-aware models are expected to exhibit the same diversity–validity trade-off observed in our formulation. While a direct empirical comparison is infeasible due to the lack of publicly available pretrained pocket-conditioned diffusion models, we expect the same behavior to hold and plan to validate this in future work.
>
> ---
>
> ## **"Runtime and compute-cost considerations for large-scale sampling."** ##
> **Appendix N** provides a detailed runtime breakdown comparing HybridLinker with standard diffusion-based linkers. As summarized in the table below, the serialized two-step pipeline, “PC-free surrogate sampling ${\rightarrow}$ PC-aware refinement”, naturally incurs additional inference time due to executing both components and the lightweight operations bridging them. However, both stages are fully parallelizable, and in large-scale generation settings (e.g., ${10k+}$ samples), the effective wall-clock cost becomes comparable to that of a single diffusion model. In practice, HybridLinker achieves substantially higher success rates while maintaining competitive throughput under parallel execution.
>
> **Table: Per-molecule inference time.**
> “Serial.” and “Para.” respectively indicate the serialized and parallelized run of two steps. “Pre.” indicates a pretrained model used in inference only. Times are in seconds (s).
>
> | **Method**                        | **Component(s)**                                 | **Time (s)** |
> |----------------------------------|--------------------------------------------------|-------------:|
> | FFLOM                            | FFLOM                                           | 0.132        |
> | DeLinker                         | DeLinker                                        | 0.240        |
> | 3DLinker                         | 3DLinker                                        | 1.080        |
> | DiffLinker                       | DiffLinker                                      | 0.565        |
> | HybridLinker (FFLOM), Serial    | DiffLinker (Pre.) + FFLOM (Pre.)                | 1.700        |
> | HybridLinker (DeLinker), Serial | DiffLinker (Pre.) + DeLinker (Pre.)             | 1.780        |
> | HybridLinker (FFLOM), Para      | DiffLinker (Pre.) + FFLOM (Pre.)                | 0.650        |
> | HybridLinker (DeLinker), Para   | DiffLinker (Pre.) + DeLinker (Pre.)             | 0.650        |

---

> ### Author Response · Authors · 2025-12-03
> **[2/2] Response to Reviewer qSuD**
>
> ## **"Dependence on the surrogate topology model and diffusion prior."** ##
>
> We agree that HybridLinker inherits characteristics from its constituent experts. To clarify its robustness, we highlight two observations. First, as shown in **Table 2**, HybridLinker achieves consistently strong performance across the two valid PC-free surrogate models, even though we use only the DiffLinker diffusion prior due to the limited availability of pretrained alternatives. Second, **Appendix K** presents a detailed analysis of how the quality of intermediate PC-free samples influences the refined outputs. HybridLinker reliably transforms energetically unstable surrogate samples into stable ones, and the energy-relaxation analysis shows a positive correlation between the magnitude of energy reduction and the initial surrogate energy. This indicates that HybridLinker can effectively process a wide spectrum of surrogate sample qualities. Taken together, these results demonstrate that HybridLinker is robust to variations in both the surrogate topology model and the quality of its intermediate samples.
>
> ---
>
> ## **"Simplified cross-domain likelihood and conditional independence assumptions."** ##
>
> We clarify that our formulation of the cross-domain likelihood—namely the simplified form and the conditional independence assumption—is both conceptually justified and empirically validated. Conceptually, within the DPS framework, a fully flexible formulation would require defining a differentiable probability density jointly over *atom types* and *coordinates*. This is not feasible in practice, and thus fixing atom types is an unavoidable and standard simplification adopted in prior work.
>
> Beyond this practical constraint, we additionally evaluated several choices for the term ${p(\tilde{E}, \tilde{R} \mid R)}$. The table below compares three designs.
>
> **(1) Physics-based likelihood:**
>
> ${p(\tilde{E}, \tilde{R} \mid R) \propto \exp\!\big(-U_{\mathrm{phys}}(\tilde{E}, \tilde{R})\big)}$
>
> ${U_{\mathrm{phys}}(\tilde{E}, \tilde{R}) = k_{\mathrm{stretch}} \sum_{i,j} \mathbb{1}_{\tilde{E}_{ij} \neq 0}\,\|\tilde{R}_i - R_i\|^2 + k_{\mathrm{angle}} \sum_{j} \|\tilde{\theta}_j - \theta_j^\star\|^2}$
>
> where ${\tilde{\theta}_j}$ and ${\theta_j^\star}$ denote the bond angles of the predicted graph ${\tilde{G}}$ and the ground-truth geometry, respectively.
>
> **(2) Simplified energy-based likelihood:**
>
> ${p(\tilde{E}, \tilde{R} \mid R) \propto \exp\!\big(-U_{\mathrm{sim}}(\tilde{E}, \tilde{R})\big)}$
>
> ${U_{\mathrm{sim}}(\tilde{E}, \tilde{R}) = \sum_{i,j} \mathbb{1}_{E_{ij}^\star \neq 0}\,\|\tilde{R}_i - \tilde{R}_j\|}$
>
> **(3) Proposed likelihood (Eq. 12):**
>
> ${p(\tilde{E}, \tilde{R} \mid R) \propto \kappa(\tilde{R}, R)\,\exp\!\big(-U_{\mathrm{sim}}(\tilde{E}, \tilde{R})\big)}$
>
> where ${\kappa}$ is a Gaussian kernel used to softly guide ${\tilde{R}}$ toward ${R}$.
>
> Across all settings, the simplified likelihood achieves the strongest empirical performance, supporting the practicality and effectiveness of this design. Moreover, adding the Gaussian kernel consistently improves coordinate quality, corroborated by the ablation results. This further justifies our chosen structured form of the likelihood ${p(\tilde{E}, \tilde{R} \mid R)}$.
>
> **Table: Ablation study on choice of likelihood. All methods use the exact match of atom types, but differ in the choice of likelihood.**
>
> | **Guidance Type**                        | **Unique** | **Novel** | **Valid** | **V+U** | **V+N** |
> |------------------------------------------|-----------:|----------:|----------:|--------:|--------:|
> | Physics-based Energy                     | 65.10      | 41.30     | 85.74     | 53.75   | 33.86   |
> | Simplified Energy                        | 67.20      | 44.10     | 69.23     | 54.59   | 35.08   |
> | Simplified Energy + Gaussian Kernel      | 68.45      | 44.69     | 69.14     | 55.10   | 35.17   |

---

### Meta-Review · Area_Chair_6b1R · 2026-01-08

**Summary:**

The novelty of proposed approached, evaluation settings are concerned.

**Reviewer Concerns:**

The reviewer pointed out that the baseline numbers are not the as the original paper. Though the authors claimed in the rebuttal that the numbers are reproduced in the new setting, it is possible that the baseline methods may also need further hyper-parameter tuning in the new setting.

**Reviewer Scores:**

No reviewer may increase their score given the current rebuttal.

---

### Decision · Program_Chairs · 2026-01-26

Reject